# Unsupervised Anomaly Detection through Mass Repulsing Optimal Transport

**Eduardo Fernandes Montesuma**[*]                                      *edumfern@gmail.com*
*Université Paris-Saclay, CEA, List, F-91120 Palaiseau France*

**Adel el Habazi**[*]                                             *adelelhabazi@gmail.com*
*École Centrale de Nantes, Nantes, France*

**Fred Ngolè Mboula**                                *fred-maurice.ngole-mboula@cea.fr*
*Université Paris-Saclay, CEA, List, F-91120 Palaiseau France*

**Reviewed on OpenReview:** *https://openreview.net/forum?id=PPGJ3EvENv*

## Abstract

Detecting anomalies in datasets is a longstanding problem in machine learning. In this context, anomalies are defined as a sample that significantly deviates from the remaining data. Meanwhile, Optimal Transport (OT) is a field of mathematics concerned with the transportation, between two probability distribution, at least effort. In classical OT, the optimal transportation strategy of a distribution to itself is the identity, i.e., each sample keeps its mass. In this paper, we tackle anomaly detection by forcing samples to displace its mass, while keeping the least effort objective. We call this new transportation problem Mass Repulsing Optimal Transport (MROT). Naturally, samples lying in low density regions of space will be forced to displace mass very far, incurring a higher transportation cost. In contrast, samples on high density regions are able to send their mass just outside an *exclusion zone*. We use these concepts to design a new anomaly score. Through a series of experiments in existing benchmarks, and fault detection problems, we show that our algorithm improves over existing methods. Our code is publicly available at ⓞ https://github.com/eddardd/MROT

## 1 Introduction

An anomaly, or an outlier, is a data point that is significantly different from the remaining data (Aggarwal, 2017), to such an extent that it was likely generated by a different mechanism (Hawkins, 1980). From the perspective of machine learning, Anomaly Detection (AD) wants to determine, from a set of examples, which ones are likely anomalies, typically through an interpretable score. This problem finds applications in many different fields, such as medicine (Salem et al., 2013), cyber-security (Siddiqui et al., 2019), and system monitoring (Isermann, 2006), to name a few. As reviewed in Han et al. (2022), existing techniques for AD are usually divided into unsupervised, semi-supervised and supervised approaches, with an increasing need for labeled data. In this paper, we focus on unsupervised AD, which does not need further labeling effort in constituting datasets.

Meanwhile, Optimal Transport (OT) is a field of mathematics concerned with the transportation of masses at least effort Villani et al. (2009). In its modern treatment, one can conceptualize transportation problems between probability distributions, which has made an important impact in machine learning research (Montesuma et al., 2024a). Hence, OT is an appealing tool, as it can be estimated non-parametrically from samples from probability distributions. Likewise, the plethora of computational tools for computing OT (Peyré & Cuturi, 2020; Flamary et al., 2021) further stresses its usability.

---

[*]These authors contributed equally to this work.

In this context, the application of OT for AD is not straightforward, as we are interested in analyzing a single probability distribution. We present a new OT problem between a distribution and itself, by restricting *where* a sample can send its mass to. More specifically, we design an *exclusion zone*, prohibiting samples from keeping its mass, or sending its mass to a small vicinity. Especially, we assume that anomalies lie in low-density regions of space, with only a few samples in their vicinity. By restricting the transport of mass in the vicinity of samples, anomalies are naturally forced to send their mass to the high-density region, which is assumed to be far away from the anomaly samples. Hence, anomalies will have an overall higher *transportation effort* than normal samples, which can find nearby samples outside the exclusion zone. We show a conceptual illustration of our method in Figure 1.

Although OT has been previously used to compare and aggregate signals in the context of AD (Alaoui-Belghiti et al., 2019; 2020), to the best of our knowledge ours is the first general purpose OT-based algorithm for AD. Furthermore, we propose a new OT problem based on the engineering of the ground-cost, which has links to OT with repulsive costs (Di Marino et al., 2017). We benchmark our algorithm in a comprehensive list of datasets, including tabular, computer vision, and natural language processing proposed by Han et al. (2022), besides fault detection (Reinartz et al., 2021; Montesuma et al., 2024b).

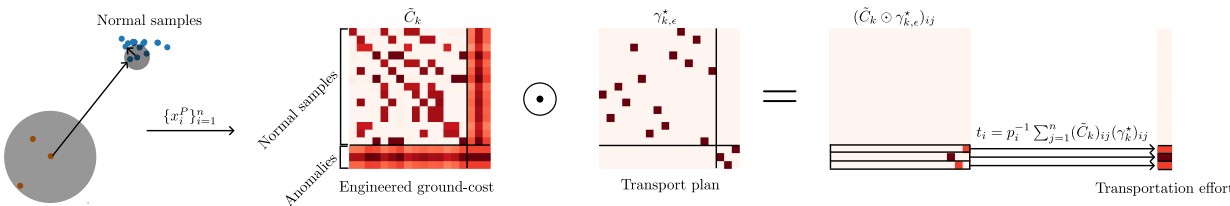

Figure 1: Mass Repulsive Optimal Transport. Our method engineers the ground-cost $c(x_i^{(P)}, x_j^{(P)})$ between samples of a distribution $P$. That way, samples are forced to send their mass outside a vicinity defined through its nearest neighbors (shaded gray areas). This design leads to a transportation plan, such that anomalous samples incur in an higher transportation effort than normal samples. We use these efforts to build an interpretable and generalizable anomaly score.

This paper is organized as follows. Section 2 discusses related work in AD and OT. Section 3 discusses our proposed method, called Mass Repulsing Optimal Transport (MROT). Section 4 covers our experiments. Finally, section 5 concludes this paper.

## 2 Related Work

**Anomaly Detection.** Following Han et al. (2022), AD methods can be mainly divided into 3 categories. First, supervised methods consider AD through the lens of binary classification under class imbalance. Second, semi-supervised methods either consider partially labeled data, or *labeled normal samples*, so that an algorithm can characterize what a normal sample is. The third, more challenging category is unsupervised AD, where the training data contains both anomalies and normal samples and labels are not available. This paper considers precisely the last setting. Next, we review ideas in unsupervised AD.

The first kind of methods rely on encoder-decode architectures to detect anomalies. The insight is that, by embedding data in a lower dimensional space, anomalies can be detected via the reconstruction error of the auto-encoding function. This is the principle of Principal Component Analysis (PCA) Shyu et al. (2003), which employs linear encoding and decoding functions, but also of kernelized versions Schölkopf et al. (1997); Hoffmann (2007), as well as neural nets Vincent et al. (2008); Bengio et al. (2013), which rely on non-linear embedding techniques.

The second type of strategies are based on the paradigm of 1-class classification. As Schölkopf et al. (1999) puts, the idea is to define a function that outputs 0 on a small, dense region of the space where normal samples lie, and 1 elsewhere. In this context, Schölkopf et al. (1997) extends the celebrated Support Vector Machine (SVM) to AD, and Liu et al. (2008) extends Random Forests (RFs) of Breiman (2001).

A third kind of approaches focuses on *neighborhoods and clustering*, to model the data underlying probability distribution, especially through the density of samples over the space. This is the case of $k-$Nearest Neighbors ($k-$NN) (Ramaswamy et al., 2000), who use distances and nearest neighbors to determine anomalies, Local Outlier Factor (LOF) Breunig et al. (2000), who devised a score that measures the local deviation of a sample with respect its neighbors. Finally, Clustering-based LOF (CBLOF) He et al. (2003) proposed an extension of LOF based on the relative sizes of clusters within the data.

**Deep Learning-based Anomaly Detection.** As Pang et al. (2021) reviews, deep learning is used for AD in mainly 3 ways: for i) learning powerful feature extractors, ii) learning representations of normality, and iii) building an end-to-end anomaly score. Besides iii, i and ii allow for the synergy between the deep and traditional AD algorithms. In fact, one can use classical AD strategies over the latent space of a neural net. We explore this possibility in Section 4.1.

In addition to this possibility, we give a few examples of how deep learning has contributed to AD. For example, Ruff et al. (2018) proposed a strategy for performing *deep* one class classification, which echoes the ideas of One Class SVM (OCSVM) Schölkopf et al. (1999).

Furthermore, generative modeling has deeply contributed to AD. For instance, generative adversarial nets (Goodfellow et al., 2014) can serve as a modeling step for capturing features of the underlying distribution $P$. For instance, in Schlegl et al. (2019) and Zenati et al. (2018), the authors combine reconstruction errors with the distance of the predictions of the discriminator for building an anomaly score. Akcay et al. (2018) takes this idea further. This strategy creates a encoder-decoder-encoder architecture for the generator network. The model is trained to model normal images, by mapping them to the latent space, reconstructing back the image, then mapping again to the latent space. Anomalous images are then detected via the usual reconstruction loss, plus the distance between latent representations and the difference between discriminator's predictions. Another direction consists of using variational inference (Dias et al., 2020) or diffusion models (Livernoche et al., 2024).

As we cover in the next section, our method uses nearest neighbors and OT to model, non-parametrically, the density of samples over the space. More specifically, we prohibit samples in OT to keep their mass, or sending it over a region of space defined through their $k-$NN. Differently from Ramaswamy et al. (2000) and Breunig et al. (2000), we do not rely on distances, which might not have a meaning in high-dimensions. Rather, we rely on the effort of transportation, measured through the samples' mass times the ground-cost.

**Optimal Transport with Repulsive Costs.** In general OT theory (see, e.g., Section 3.1 below), samples are transported based on a ground-cost that measures how expensive it is to move masses between measures. In its original conception by Monge (1781) and Kantorovich (1942), this ground cost is the Euclidean distance $c(\mathbf{x}_1, \mathbf{x}_2) = \|\mathbf{x}_1 - \mathbf{x}_2\|_2^2$. As reviewed in Di Marino et al. (2017), it may be interesting to consider *repulsive costs*, i.e. functions $c$ that are big when $\mathbf{x}_1$ and $\mathbf{x}_2$ are close to each other and small otherwise. An example of such costs, arising from physics, is the Coulomb interaction $c(\mathbf{x}_1, \mathbf{x}_2) = (\|\mathbf{x}_1 - \mathbf{x}_2\|_2)^{-1}$. Still following Di Marino et al. (2017), these kinds of transportation problems proved useful in physics, e.g., for quantum mechanics and $N-$body systems.

In this paper, we consider a different kind of repulsive cost, which we call the mass repulsive cost (see, e.g., Section 3.2 below). Our notion of cost defines an exclusion zone, based on its nearest neighbors, where sending mass is too costly. As a result, our approach captures the local characteristics of the probability distribution being analyzed, especially its density. A special characteristic of our approach is to give a sense of the transportation from a distribution to itself.

**Optimal Transport-based Anomaly Detection.** Previous works (Alaoui-Belghiti et al., 2019; 2020) have considered OT for AD. These works proposed a distance-based detection mechanism, in which isolated samples are considered anomalies. OT contributes to this setting, by defining a rich metric between samples. Especially, these works considered AD in time series data, and OT is used to compute distances between those time series in the frequency domain, under a Chebyshev ground cost. In comparison with these methods, ours is notably general purpose, that is, we do not assume data to be time series. Instead of using a Chebyshev ground cost, we model the AD problem with a repulsive cost.

# 3 Proposed Method

## 3.1 Optimal Transport

OT is a field of mathematics, concerned with the displacement of mass between a source measure, and a target measure, at least effort. In the following, we cover the principles of OT in continuous and discrete settings. We refer readers to Peyré & Cuturi (2020) for a computational exposition of the main concepts, and Montesuma et al. (2024a) for applications in machine learning. In the following, we are particularly interested in the formulation by Kantorovich (1942), which is defined as,

**Definition 3.1.** *(Kantorovich Formulation) Let $P$ and $Q$ be 2 probability distributions over a set $\mathcal{X}$. Let $c : \mathcal{X} \times \mathcal{X} \to \mathbb{R}$ be a* ground-cost*, measuring the effort of transporting units of mass from $x$ to $y$. Let $\Gamma(P,Q) = \{\gamma \in \mathbb{P}(\mathcal{X} \times \mathcal{X}) : \int_{\mathcal{X}} \gamma(x,B)dx = Q(B) \text{ and } \int_{\mathcal{X}} \gamma(A,y)dy = P(A)\}$ be the set of* transportation plans*, whose marginals are $P$ and $Q$. The optimal transportation problem is written as,*

$$\gamma^\star = OT(P,Q) = \underset{\gamma \in \Gamma(P,Q)}{arg \ inf} \int_{\mathcal{X} \times \mathcal{X}} c(x,y)d\gamma(x,y). \tag{1}$$

Equation 1 defines the transportation problem as an infinite dimensional linear program on the variable $\gamma$, called transport plan. In our case, instead of having access to a closed-form $P$, one has samples $\{\mathbf{x}_i^{(P)}\}_{i=1}^n$, each $\mathbf{x}_i^{(P)} \sim P$ with probability $p_i$. In such cases, $P$ may be approximated with an empirical measure,

$$\hat{P}(\mathbf{x}) = \sum_{i=1}^n p_i \delta(\mathbf{x} - \mathbf{x}_i^{(P)}), \sum_{i=1}^n p_i = 1, p_i \geq 1, \forall i. \tag{2}$$

Plugging back equation 2 into equation 1 leads to a finite linear program,

$$\hat{\gamma} = \underset{\gamma \in \Gamma(\mathbf{p}, \mathbf{q})}{arg \ min} \langle \gamma, \mathbf{C} \rangle_F = \sum_{i=1}^n \sum_{j=1}^m \gamma_{ij} \underbrace{c(\mathbf{x}_i^{(P)}, \mathbf{x}_j^{(Q)})}_{C_{ij}}, \ \Gamma(\mathbf{p}, \mathbf{q}) = \{\gamma : \sum_i \gamma_{ij} = q_j, \sum_i \gamma_{ij} = p_i \text{ and } \gamma_{ij} \geq 0\} \tag{3}$$

where the optimization variables are the coefficients $\gamma_{ij}$ of the transport plan. Problem 3 is a finite linear program, hence the solution $\gamma$ is a sparse matrix with at most $n + m - 1$ non-zero elements (Peyré & Cuturi, 2020). Solving it through the celebrated Simplex algorithm (Dantzig, 1983), which has computational complexity $\mathcal{O}(n^3 \log n)$ and storage complexity $\mathcal{O}(n^2)$ (i.e., storing each $\gamma_{ij}$). We show an example of empirical OT in Figure 2.

A faster alternative was introduced by Cuturi (2013), who shown that adding an entropic regularization to equation 3 leads to a problem that can be solved through Sinkhorn's algorithm (Sinkhorn, 1967). From a continuous perspective, this is equivalent to penalizing Kullback-Leibler (KL) divergence between $\gamma$, and the trivial coupling $P \otimes Q = P(x)Q(y)$. This regularization term is related to the entropy of $\gamma$ as discussed in (Peyré & Cuturi, 2020, Chapter 4), hence this problem is called entropic OT. Next, we define the entropic OT problem,

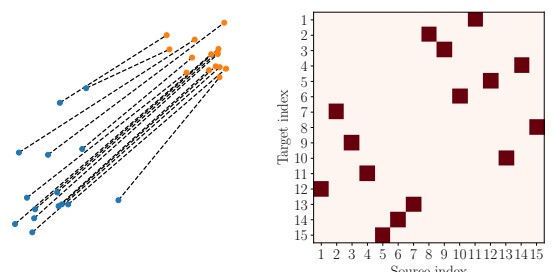

Figure 2: Optimal transport plan $\gamma$ between samples of $P$ (blue) and $Q$ (orange). On the left, we connect samples $(i,j)$ for which $\gamma_{ij}^\star > 0$. On the right, we show the entries of $\gamma^\star$.

**Definition 3.2.** *(Entropic Optimal Transport) Under the same conditions of Definition 3.1, let $\epsilon \geq 0$ be an entropic penalty. The entropic OT problem is given by,*

$$\gamma_\epsilon^\star = OT_\epsilon(P,Q) = \underset{\gamma \in \Gamma(P,Q)}{arg \ inf} \int_{\mathcal{X} \times \mathcal{X}} c(x,y)d\gamma(x,y) + \epsilon KL(\gamma | P \otimes Q), \tag{4}$$

where $KL(\gamma|\xi) = \int_{\mathcal{X}\times\mathcal{X}} \log\left(\frac{d\gamma}{d\xi}(x,y)\right) d\gamma(x,y) + \int_{\mathcal{X}\times\mathcal{X}} (d\xi(x,y) - d\gamma(x,y))$ *is the KL divergence between measures $\gamma$ and $\xi$ (Peyré & Cuturi, 2020, Eq. 4.10).*

We can obtain an equivalent discrete formulation by plugging back equation 2 into 4, which leads to,

$$\hat{\gamma}_\epsilon = \arg\min_{\gamma\in\Gamma(\mathbf{p},\mathbf{q})} \langle\gamma, \mathbf{C}\rangle_F - \epsilon H(\gamma) = \sum_{i=1}^{n}\sum_{j=1}^{m} \gamma_{ij}C_{ij} + \epsilon\sum_{i=1}^{n}\sum_{j=1}^{m} \gamma_{ij}(\log\gamma_{ij} - 1), \tag{5}$$

where $H(\gamma)$ denotes the entropy of the transportation plan. Since equation 5 relies on the Sinkhorn (1967) algorithm rather than linear programming, it has $\mathcal{O}(Ln^2)$ computational complexity, where $L$ is the number of iterations. In general, the KL and entropic terms in equations 4 and 5 have a smoothing effect on the transportation plan $\gamma_\epsilon$. As a result, $\hat{\gamma}_\epsilon$ has more non-zero elements than $\hat{\gamma}$.

In the next section, we explore a new transportation problem with a single probability distribution. This problem is understood as the transportation of $P$ to itself, when the samples form $P$ are forced to send their mass outside of their immediate neighborhood.

## 3.2 Mass Repulsing Optimal Transport

In this section, we propose a new OT problem called MROT. This problem is inspired by the Kantorovich formulation, described in section 3.1. However, instead of considering two different probability distributions $P, Q$, it considers the transport of $P$ to itself. Due to the properties of OT, if we consider the OT plan $\gamma^\star = OT(P, P)$, it is supported in the set $\{(x,x), x \in \mathcal{X}\}$, that is, each point keeps its own mass (Santambrogio, 2015). This motivates our new problem, in which we force points to *repell* its mass. Henceforth, we assume that $c$ comes from a metric $(\mathcal{X}, d)$, i.e., $c(x,y) = d(x,y)^p$, $p \in [1, +\infty)$. For pairs $x \in \mathcal{X}$, $y \in \mathcal{X}$, and $L \in [0, +\infty)$,

$$\tilde{c}(x,y) = \begin{cases} c(x,y) & \text{if } y \notin \mathcal{N}(x), \\ L & \text{otherwise,} \end{cases} \tag{6}$$

where $\mathcal{N}(x)$ denotes the vicinity of $x$ (e.g., $k-$nearest neighbors, or an $\rho-$ball centered at $x$). In the next remark, we give some geometric intuition behind the cost engineering, as well as some motivation for the choice of $L$. More generally, in the next section we give a theoretically motivated choice for $L$.

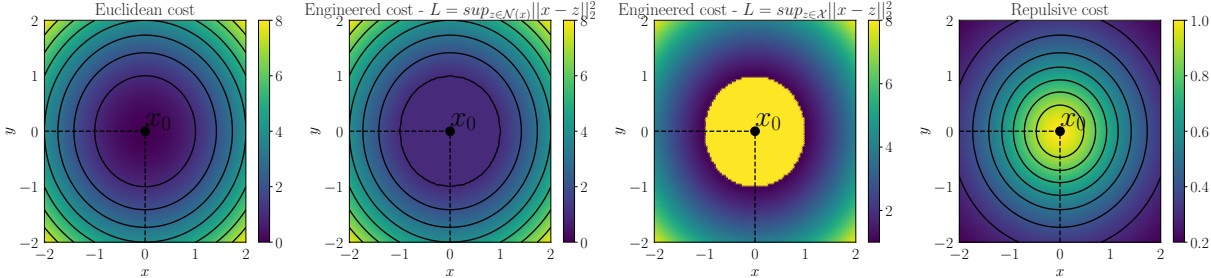

Figure 3: Comparison between different ground costs for $x_0 = (0,0)$. From left to right: Squared euclidean cost $c(x, x_0) = \|x - x_0\|_2^2$, Engineered cost (c.f., equation 6) for two choices of $L$, and repulsive cost.

**Remark 3.1.** *(Geometric intuition) Assume $x_0 \in \mathcal{X} = [-2,2]^2$ fixed. Our cost engineering strategy replaces the base ground cost $c(x_0, y)$, in the neighborhood $\mathcal{N}(x_0)$ with a value $L$. Assume, for argumentation purposes, that $L \to +\infty$. This choice means that it is infinitely costly for each point to keep its own mass. As a result, samples are encouraged to send their mass to the immediate outside of their neighborhood – thus, we call this idea* mass repulsive OT.

*To ground some intuition, we show, in Figure 3, a comparison using the squared Euclidean distance as the ground-cost, i.e., $c(x,y) = \|x - y\|_2^2$. Alongside the base cost, we show our engineered cost, and the repulsive*

*cost $(1 + c(x, y))^{-1}$. Here, we define the neighborhood of $x_0$ through the $\rho-$closed ball centered at $x_0$, i.e., $\mathcal{N}(x_0) = \{y : \mathbb{R}^2 : \|x - y\|_2 \leq \rho\}$. Likewise, we use $\rho = 1$, and, $L = sup_{z \in \mathcal{N}(x_0)} c(x, z)$. This particular choice leads to nice theoretical properties for the continuous MROT problem (see equation 7 below).*

*Due to the particular choices we made, the supremum takes its value on the frontier $\partial\mathcal{N}(x_0) = \{z : \mathbb{R}^2 : \|x_0 - z\|_2 = \rho\}$, in which case $c(x_0, z) = \rho^2$, independently of $x_0$. This choice for $L$ effectively flattens the ground cost around the vicinity of $x_0$. As we show in the next section, this is essential for ensuring the existence of the continuous MROT problem. In practice, it is beneficial to set $L$ to a large, finite constant to ensure points are transported outside $\mathcal{N}(x_0)$. Note that, as we remark in the next section, this does not harm the existence of the **discrete** MROT solution. We show an example of this choice by setting $L = sup_{z \in \mathcal{X}} \|x - z\|_2^2$.*

Our principle of mass repulsion is different from repulsive costs (Di Marino et al., 2017), which are designed to model the interaction between particles in multimarginal OT (Pass, 2015). Indeed, we design a transportation problem from a probability distribution to itself. Hence, while repulsive costs incentive transportation towards distant points in space, our mass repulsing cost induces transportation just outside an exclusion zone. Henceforth, we focus on $p = 2$ for the Euclidean distance. The ground-cost we propose essentially defines an *exclusion zone* around points **x**, where these points are discouraged from sending their mass.

Our main hypothesis for anomaly detection is that anomalous points lie in low density regions of $P$. On the one hand, If these points are forced to send its mass outside its vicinity, it will be forced to send it to high density regions of $P$ – otherwise, mass conservation in OT would not hold. On the other hand, *if the exclusion zone is smaller than high density regions of $P$*, points on these regions will be sent close-by. As a result, anomalous points will have a higher transportation cost (c.f., equations 9 and 10 below) than normal points. We thus consider the following OT problem,

$$\gamma_\epsilon^\star = \text{MROT}_\epsilon(P) = \inf_{\gamma \in \Gamma(P,P)} \int \tilde{c}(x, y) d\gamma(x, y) + \epsilon \text{KL}(\gamma | P \otimes P), \tag{7}$$

which, like the continuous entropic OT problem in definition 3.2, also admits a discrete version when $P$ is approximated empirically through equation 2,

$$\gamma_\epsilon^\star = \text{MROT}_\epsilon(\hat{P}) = \min_{\gamma \in \Gamma(\mathbf{p}, \mathbf{p})} \langle \gamma, \tilde{\mathbf{C}} \rangle_F - \epsilon H(\gamma). \tag{8}$$

Considering again Figure 3, we focus on comparing the commonly used squared Euclidean cost, our engineered cost, and a repulsive cost associated with the Coulomb interaction (Di Marino et al., 2017). In traditional OT, sending mass to distant regions of space is costly. As a result, points are encouraged to keep their own mass as close as possible. Consequently, OT from $P$ to itself is the trivial plan $\gamma = \text{Id}$.

For our engineered cost, we use $\mathcal{N}(\mathbf{x}) = \{\mathbf{y} : \|\mathbf{x} - \mathbf{y}\|_2^2 \leq \rho\}$, $\rho = 1$. Both our engineered cost $\tilde{c}$ and the repulsive cost $(1 - c(x, x_0))^{-1}$ assign high costs to points near $x_0$. In particular, within the neighborhood $\mathcal{N}(x_0)$, our engineered cost $\tilde{c}(x, x_0)$ reaches its maximum. Hence, there is an important difference between MROT and OT with repulsive costs. Our engineered cost promotes transportation *to the immediate region outside $\mathcal{N}$*, whereas the repulsive cost drives samples as far as possible from $x_0$. As we discussed previously, this feature of the ground-cost allows us to build an anomaly score.

### 3.3 Theoretical Analysis

In this section, we provide a proof for the existence of MROT plans. While at least one solution of equation 7 exists, its uniqueness is not, in general, guaranteed. In this section, we denote the set of measures over a set $\mathcal{X}$ by $\mathcal{P}(\mathcal{X})$, and the power-set of $\mathcal{X}$ by $\mathbb{P}(\mathcal{X})$.

From classical OT theory, the Kantorovich problem (c.f., equation 1) has a solution under mild conditions on the ambient space $\mathcal{X}$ and the ground-cost $c$, see Santambrogio (2015). Indeed, $\mathcal{X}$ is required to be Polish (i.e., complete, separable metric space) and $c$ must be lower semi-continuous. For simplicity, we state our results on $d-$dimensional Euclidean spaces, which are the setting for our experiments. We start by re-stating (Santambrogio, 2015, Theorem 1.5),

**Theorem 3.1.** *(Existence of Optimal Transport plans) Let $\mathcal{X}$ be a Polish space, $P$, and $Q \in \mathcal{P}(\mathcal{X})$, and $c : \mathcal{X} \times \mathcal{X} \to [0, +\infty]$ be lower semi-continuous. Then equation 1 admits a solution.*

Since we are dealing with $d-$dimensional Euclidean spaces, the condition on the ambient space is already satisfied. It remains the question of the lower semi-continuity of our engineered cost. In the next theorem, we prove that, for the squared Euclidean cost, a solution for MROT exists,

**Theorem 3.2.** *Let $c(x,y) = \|x - y\|_2^2$ and $\mathcal{N}(x) = \{y \in \mathbb{R}^d : \|x - y\|_2 \leq \rho\}$. Then the continuous MROT problem (equation 7) admits a solution.*

*Proof.* Let $x_0 \in \mathbb{R}^d$ and $\rho > 0$ be given. Our proof relies on the decomposition of $y \in \mathbb{R}^d$ in 3 regions: i) $\|x_0 - y\|_2 > \rho$, ii) $\|x_0 - y\|_2 < \rho$, and iii) $\|x - y\|_2 = \rho$. For i) and ii), $\tilde{c}$ is a continuous function of its inputs. Indeed, for $\|x_0 - y\|_2 > \rho$, $\tilde{c}(x,y) = \|x_0 - y\|_2^2$. Likewise, for $\|x_0 - y\|_2 < \rho$, we have $\tilde{c}(x,y) = \rho^2$. Now, we need to prove that $\tilde{c}$ is continuous on $\|x_0 - y\|_2 = \rho$. We sub-divide this into 2 cases. First, assume we have a sequence $(y_n)$, $y_n \to y$, $y_n \notin \mathcal{N}(x)$. Since $y_n \notin \mathcal{N}(x_0)$, we have $\tilde{c}(x_0, y_n) = \|x_0 - y_n\|_2^2$. From the continuity of the squared Euclidean distance, $\tilde{c}(x_0, y_n) \to \|x_0 - y\|_2^2 = \rho^2$. Second, assume we have a sequence $(y'_n)$, $y'_n \to y$, $y'_n \in \mathcal{N}(x_0)$. Then, $\tilde{c}(x_0, y'_n) = \rho^2 \to \tilde{c}(x_0, y) = \rho^2$. Hence, $\tilde{c}$ is continuous. $\qquad\square$

The principles used in the previous theorem can be generalized to other costs and neighborhoods, as long as: i) the base cost $c(x,y)$ is continuous, ii) the neighborhood $\mathcal{N}(x)$ is compact, and iii) the supremum $\sup_{z \in \mathcal{N}(x)} c(x,z)$ is attained in the boundary of $\mathcal{N}(x)$, $\forall x \in \mathcal{X}$. The bottom line of our analysis is that, in general, one needs to carefully engineer the ground cost, through the choice of the base ground cost $c$, and the neighborhood function $\mathcal{N}$.

Although existence may hold, the fact that the engineered cost introduces a flat region on the neighborhood $\mathcal{N}(x)$ of $x$ makes it difficult to establish the uniqueness of the MROT plan. We recall from (Figalli & Glaudo, 2021, Section 2.6) that, in classical OT, the existence of a unique transport plan is related to the existence of a Monge map between $P$ and $Q$. From (Figalli & Glaudo, 2021, 2.7.1), this means that we need 3 conditions: i) $x \mapsto \tilde{c}(x,y)$ is differentiable, ii) $y \mapsto \nabla_x \tilde{c}(x,y)$ is injective, and iii) for $\rho > 0$, and $B_\rho = \{x \in \mathcal{X} : \|x\| \leq \rho\}$, $|\nabla \tilde{c}(x,y)| \leq C_\rho$ for every $x \in B_\rho$. As it turns out, neither of these conditions holds for our engineered cost, so the MROT plan is likely not unique.

Now, let us focus on the discrete case, which is of practical interest to us. Note that $\Gamma(\mathbf{p}, \mathbf{p})$ is non-empty as $\gamma_{ij} = p_i p_j \in \Gamma(\mathbf{p}, \mathbf{p})$. Furthermore, this set is compact, and the objective in equation 8 is continuous. Then, by Weierstrass theorem (Santambrogio, 2015, Box 1.1), there exists a minimizer to the discrete setting. Here, note that we did not have to impose continuity or lower semi-continuity on the neighborhood function $\mathcal{N}$ or the ground cost $\tilde{c}$. As in classical OT, the solution to this problem is not unique, since the objective function is not strictly convex. Following this line, adding the entropic penalty (i.e., $\epsilon > 0$ in equation 8) makes the objective strictly convex (Peyré & Cuturi, 2020, Section 4.1), which guarantees the uniqueness of the OT plan, and hence, the uniqueness of the MROT plan.

### 3.4 Building and Generalizing an Anomaly Score

Through MROT, we want to build a score for the samples based on how anomalous they are. Assuming that anomalies lie in a low density region of space, our MROT problem forces those samples to send their mass to distant parts of the feature space. In contrast, normal samples can send their mass to the immediate neighborhood outside the exclusion zone. As a result, we can sort out normal from anomalous samples using the transportation effort,

$$\mathcal{T}(x) = \mathbb{E}_{y \sim \gamma(\cdot | x)} [\tilde{c}_k(x,y)] = \int_{\mathcal{X}} \tilde{c}_k(x,y) d\gamma(y|x), \qquad (9)$$

where $\gamma(y|x)$ corresponds to the conditional probability, calculated through the joint $\gamma(x,y)$, given $x$. For empirical measures, this quantity can be calculated as follows,

$$t_i = \mathcal{T}(x_i^{(P)}) = \sum_{j=1}^{n} \frac{\gamma_{ij}^{\star}}{p_i} \tilde{c}_k(x_i^{(P)}, x_j^{(P)}), \qquad (10)$$

where $p_i$ is the importance of the $i-$th sample (e.g, $p_i = n^{-1}$ for uniform importance). Interestingly, equation 10 is similar to the barycentric map, widely used in domain adaptation Courty et al. (2016). $\mathcal{T}$ has 2 shortcomings as an anomaly score. First, it is hardly interpretable, as its range depends on the choice of ground-cost $c$. Second, it is only defined in the support of $\hat{P}$. We offer a solution to both of these problems.

Concerning interpretability, we propose to transform it using the Cumulative Distribution Function (CDF) of its values. Let $P_{\mathcal{T}}$ be the probability distribution associated with $\mathcal{T}(x)$. The CDF is simply $F_{\mathcal{T}}(t) = P_{\mathcal{T}}((-\infty, t))$. Naturally, since $P_{\mathcal{T}}$ is not available, it may be approximated from samples $\{t_i\}_{i=1}^n$, obtained through equation 10. We do so through, Kernel Density Estimation (KDE),

$$\hat{P}_{\mathcal{T}}(t) = \frac{1}{n\sigma}\sum_{i=1}^n \phi\left(\frac{t-t_i}{\sigma}\right), \text{ and, } \hat{F}_{\mathcal{T}}(t) = \int_{-\infty}^t \hat{P}_{\mathcal{T}}(s)ds \tag{11}$$

where $\phi$ is a kernel function (e.g., the Gaussian kernel $\phi(x) = \exp(-x^2/2)$), and $\sigma$ is the bandwidth, controlling the smoothness of $\hat{P}_{\mathcal{T}}$ and determined through Scott's rule (Scott, 1979). Equation 11 is an approximation for the density of transportation efforts $\{t_i\}_{i=1}^n$. The CDF is appealing, as it is a monotonic function over transportation effort $t \in \mathbb{R}$, and it takes values on $[0, 1]$, both of which are desirable for an anomaly score.

The question of how to *extrapolate* the anomaly score for new samples remains. For example, even if we use the CDF values as anomaly scores, we need to recalculate $t = \hat{\mathcal{T}}(\mathbf{x})$ for a new sample $\mathbf{x} \sim P$, which is challenging, as $\hat{\mathcal{T}}$ is only defined on the support of $\hat{P}$. A naive approach would be to append $\mathbf{x}$ to the set $\{\mathbf{x}_i^{(P)}\}_{i=1}^n$ and solve a MROT problem again. Naturally, this is not feasible, as solving an OT problem for each new sample is computationally expensive.

In this paper, we present the more efficient idea of modeling the relationship $\mathbf{x} \mapsto \hat{F}_{\mathcal{T}}(\hat{\mathcal{T}}(\mathbf{x}))$ from the samples we receive, that is, we create a labeled data set $\{\mathbf{x}_i^{(P)}, t_i\}_{i=1}^n$, where $t_i = \hat{\mathcal{T}}(\mathbf{x}_i^{(P)})$. Our anomaly score comes, then, through a function $\psi(\mathbf{x}_i^{(P)})$ fit to the labeled dataset through regression. The fitting $\psi$ can be done with standard regression tools, such as Ordinary Least Squares (OLS), Support Vector Regression (SVR) (Smola & Schölkopf, 2004), nearest neighbors or gradient boosting (Friedman, 2002). In general, one can expect the relationship between $\mathbf{x}_i^{(P)}$ and $\hat{F}_{\mathcal{T}}(\hat{\mathcal{T}}(\mathbf{x}_i^{(P)}))$ to be non-linear, hence, it is generally necessary to use a non-linear regression model. We show a summary of our strategy in Algorithm 1.

---

**Algorithm 1:** Mass Repulsive Optimal Transport.

---

**1** function mrot($\mathbf{X}^{(P)}, \epsilon$)

**2**    $\gamma_{k,\epsilon}^\star = \text{MROT}_{k,\epsilon}(\hat{P})$;

**3**    $t_i \leftarrow \sum_{j=1}^n ((\gamma_\epsilon^\star)_{ij}/p_i)\tilde{c}(\mathbf{x}_i^{(P)}, \mathbf{x}_j^{(P)})$;

**4**    $\hat{P}_{\mathcal{T}} \leftarrow \text{KDE}(\{t_i\}_{i=1}^n)$;

**5**    $\psi \leftarrow \text{Regression}(\{\mathbf{x}_i^{(P)}, \hat{F}_{\mathcal{T}}(t_i)\}_{i=1}^n)$;

**6**    **return** $\psi$;

---

## 4 Experiments

We divide our experiments in 3 parts. Section 4.1 shows our results on AdBench (Han et al., 2022). Section 4.2 shows our experiments in fault detection on the Tennessee Eastman Process (Montesuma et al., 2024b; Reinartz et al., 2021). Finally, Section 4.3 explores the robustness of our methods to various hyper-parameters and design choices.

In the following, we consider adaptive neighborhoods $\mathcal{N}_k(\mathbf{x})$ which take the $k-$nearest neighborhood of $\mathbf{x}$, and set $C_{ij} = \max_{\ell=1,\cdots,n}C_{i\ell}$, where $j \in \mathcal{N}_k(\mathbf{x}_i^{(P)})$. Before going through our experiments, we show a toy example, going through all the steps in our algorithm. See Figure 3 for a comparison.

**An introductory example.** Before comparing our method with prior art, we give an introductory example that illustrates how we create anomaly scores out of samples. In this example, we sample normal examples $\mathbf{x}_i \sim \mathcal{N}(\mathbf{0}, 0.25\mathbf{I}_2)$, and anomalous samples $\mathbf{y}_j \sim \mathcal{N}([-3, -3], 0.01\mathbf{I}_2)$, where $\mathbf{I}_2$ is a $2 \times 2$ identity matrix. The dataset for this toy example consists of the concatenation $\mathbf{X}^{(P)} = \{\mathbf{x}_1, \cdots, \mathbf{x}_n, \mathbf{y}_1, \cdots, \mathbf{y}_m\}$, where $n = 500$

and $m = 25$, which means that roughly 5% of the total number of samples are anomalies. These samples are shown in Figure 5.

We compare our engineered cost with a regularized Coulomb interaction, $C_{ij} = (1 + \|\mathbf{x}_i^{(P)} - \mathbf{x}_j^{(P)}\|_2)^{-1}$. This cost is shown alongside our proposed engineered cost in Figure 4. *Note that, in Figures 5 (a - c) the lower right corner of the matrices corresponds to anomalies.*

From Figure 5, OT with the Euclidean cost results in a trivial OT solution, because the diagonal entries $C_{ij}$ are zero. Hence, $\gamma^\star = \mathbf{I}_n$ achieves 0 transportation cost. This is not the case for our engineered cost, and the regularized Coulomb interaction, shown in Figures 5 (c) and (d). To emphasize this idea, we show in Figure 6 the transportation plans acquired by MROT, and the regularized Coulomb cost. For the Coulomb cost, the anomalies send and receive less mass than normal samples. This is somewhat expected, as the anomalous samples are clustered together in a tight region of $\mathbb{R}^2$, resulting in a high cost. As a result, these samples are encouraged to send their mass elsewhere. This phenomenon does not happen with our engineered cost, which encourages samples to send their mass *just outside* to their exclusion zone.

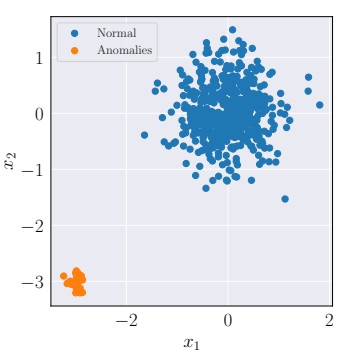

Figure 4: Toy example.

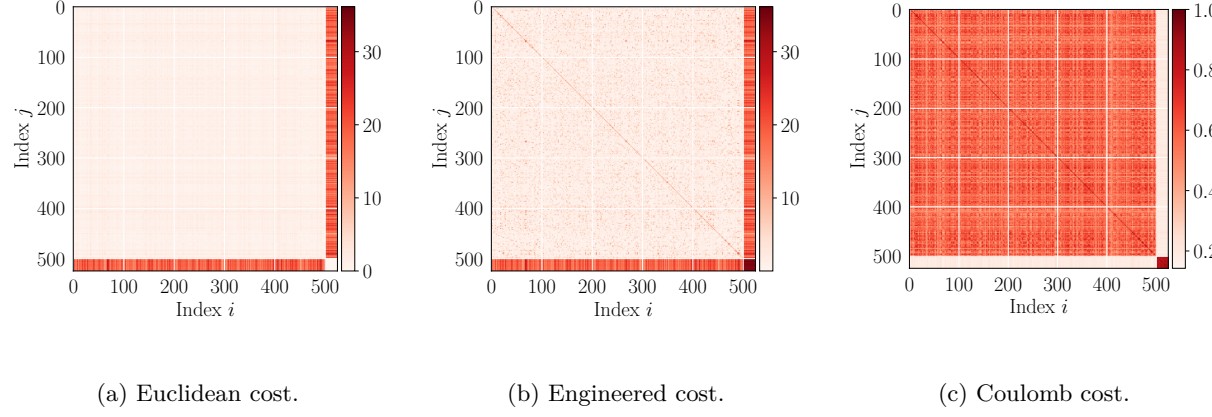

| (a) Euclidean cost. | (b) Engineered cost. | (c) Coulomb cost. |

Figure 5: In (a), (b) and (c), we show the ground-cost $C_{ij}$ between samples in Figure 4, using the Euclidean distance, our engineered ground-cost (equation 6), and the regularized Coulomb interaction cost.

Hence, there is an important distinction between our approach and using repulsive costs in OT. Considering our last remark about the regularized Coulomb interaction, samples send their mass to distant regions of space, thus reducing $C_{ij}$. As a result, *anomalies will incur in a smaller transportation cost than normal samples.* Although counterintuitive, one can still construct a detection rule out of this idea.

In Figure 7, we present the anomaly scores derived from our MROT strategy and the regularized Coulomb cost. These scores are computed by first determining the transportation effort (c.f., equation 10) for each sample $x_i^{(P)}$, then estimating the density of these efforts, and finally transforming the density into a $[0, 1]$ score via its CDF, as outlined in section 3.4. As shown in Figure 7, the MROT approach assigns higher anomaly scores to anomalous samples compared to normal ones, simplifying the process of setting a threshold for anomaly detection. In contrast, the regularized Coulomb cost exhibits the opposite behavior. Indeed, anomalous samples send their mass to distant parts of space (i.e., to normal samples), which, due the nature of the Coulomb cost, lead to a smaller transportation effort. Nevertheless, as discussed previously, it is still possible to establish a detection rule, albeit being counterintuitive.

As we discussed throughout section 3.4, the scores in Figure 7 are only defined in the support of $\hat{P}$. We then explore the regression of this score through regression, which can be done with a variety of standard

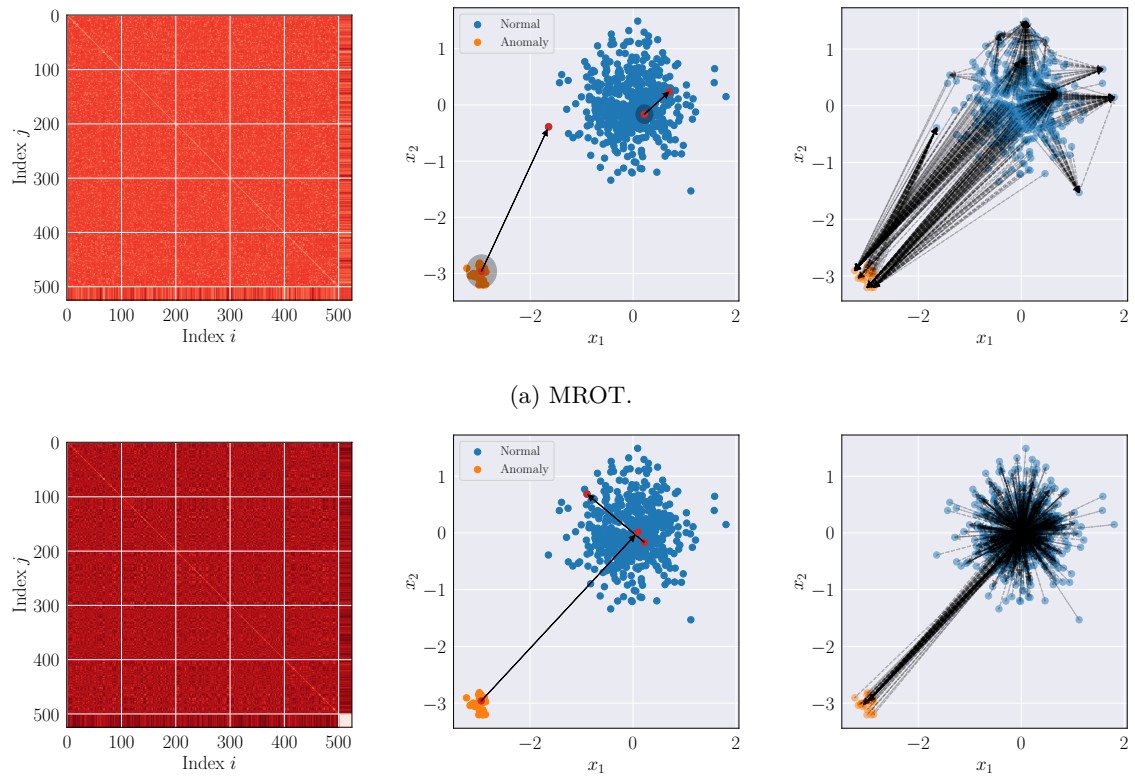

(a) MROT.

(b) OT with the regularized Coulomb cost.

Figure 6: In (a), we show the MROT transportation plan (left side), the matching (center) and the transport plan (right side). Likewise, in (b) we show the transportation plan and strategy for the regularized Coulomb cost. As shown in (b) left side and right side, the anomalies (lower right corner) sends and receives much less mass than other normal samples.

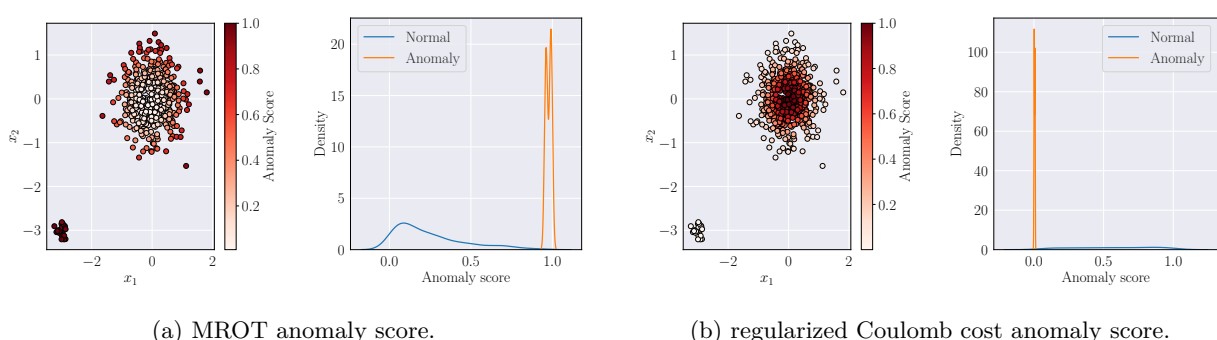

(a) MROT anomaly score.

(b) regularized Coulomb cost anomaly score.

Figure 7: Anomaly score comparison between the MROT (ours, a) and OT with repulsive costs (b). Scores are computed by calculating the transportation effort (Eq. 10) for each sample $x_i^{(P)}$, estimating their density, and normalizing to the $[0, 1]$ range via the CDF (c.f., section 3.4). Overall, our strategy assigns higher scores (close to 1.0) to anomalous samples, while assigning smaller scores to normal samples.

regression algorithms. In Figure 8 we show the results for eXtreme Gradient Boosting (XGBoost) (Chen & Guestrin, 2016) and SVR (Smola & Schölkopf, 2004), which both define the anomaly score over the whole ambient space $\mathbb{R}^2$. As an important remark, the relationship between $\mathbf{x}_i^{(P)}$ and its score is likely non-linear. Indeed, this idea is evidenced in Figure 8. As a result, one needs a non-linear regression algorithm.

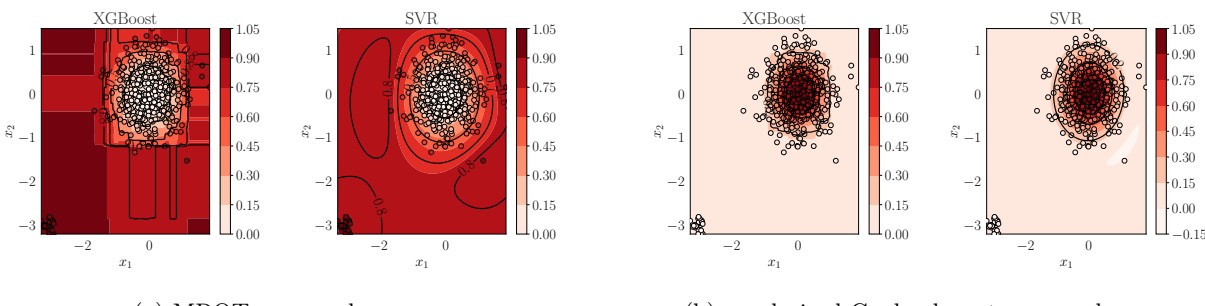

(a) MROT regressed score.                    (b) regularized Coulomb cost regressed score.

Figure 8: Regression of the anomaly score for MROT (ours, a) and OT with repulsive costs (b). While our method attributes higher scores to anomalous regions of the space, OT with repulsive costs does the inverse.

## 4.1 Comparison on AdBench

AdBench (Han et al., 2022) is a benchmark in AD with 57 different kinds of datasets, grouped into 47, 5 and 5 real-world, vision and Natural Language Processing (NLP) datasets. In our experiments, we focus on real-world and natural-language datasets. We compare, in total, 17 methods, grouped into classical, diffusion-based and our proposed methods based on OT. For classic methods, we consider Isolation Forest (IsoF) (Liu et al., 2008), OCSVM (Schölkopf et al., 1999), $k-$NN, LOF (Breunig et al., 2000), CBLOF (He et al., 2003), Empirical Cumulative Distribution Functions (ECOD) (Li et al., 2022), Copula-Based OD (COPOD) (Li et al., 2020), Lightweight On-line Detector of Anomalies (LODA) (Pevnỳ, 2016), Feature Bagging (Lazarevic & Kumar, 2005), and Histogram-based Outlier score (HBOS) (Goldstein & Dengel, 2012). For diffusion based, we consider the 4 variants of Diffusion Time Estimation (DTE) (Livernoche et al., 2024). We also consider OT with repulsive costs Di Marino et al. (2017), and MROT (ours).

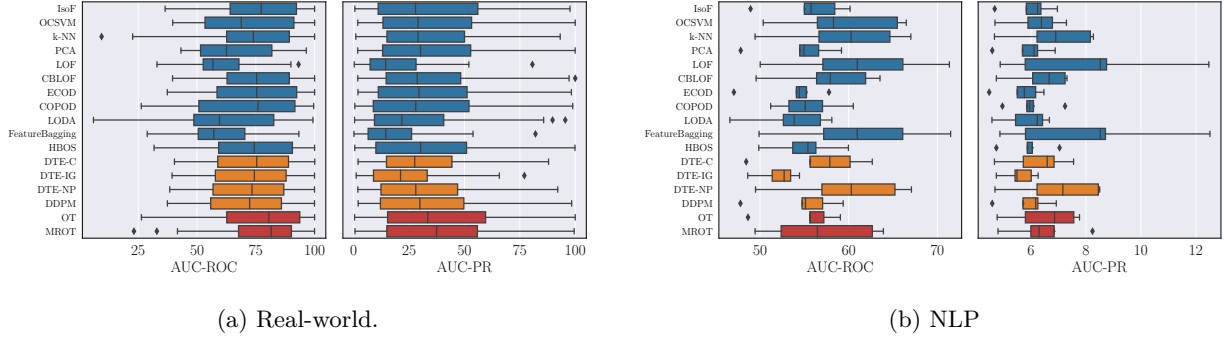

(a) Real-world.                                    (b) NLP

Figure 9: AdBench result summary. AUC-ROC and AUC-PR per dataset is available in the appendix. We compare, in total, 17 algorithms over 47 real-world datasets (a) and 5 NLP datasets (b). For real-world datasets, our MROT has state-of-the-art performance, especially when compared with recently proposed flow-based models. For higher dimensional data, our method

We show our summarized results in Figures 9 (a) and (b), for real-world and NLP datasets, respectively. First, we note that MROT and OT with repulsive costs have superior performance with respect other methods on real-world datasets. This result highlights the usefulness of OT theory in the analysis of probability distributions. Furthermore, on average, our MROT has better performance than OT (77.36% versus 75.97% ROC-AUC), proving the effectiveness of our cost engineering strategy. We refer readers to our appendix for detailed results per dataset.

Here, we highlight two limitations of our method. The first limitation of using MROT comes from the scalability of OT. As a linear program, it has at least $\mathcal{O}(n^2)$ storage, and $\mathcal{O}(n^3 \log n)$ computational complexity, where $n$ is the number of samples. In our experiments, we limited the number of samples to $n = 20,000$, by

down-sampling larger datasets. As reported in Figure 9 (a), and our detailed results given in the Appendix, this process does not affect performance. Furthermore, from Figure 9 (b), we see that our method struggles in high dimensional AD, such as those in the NLP datasets of Han et al. (2022). This limitation stems from the use of OT in high-dimensions (Montesuma et al., 2024a, Section 8.1).

## 4.2 Tennessee Eastman Process

In this section, we focus in comparing anomaly detection methods for fault detection in control systems. In this context, as defined by Isermann (2006), a fault is an anomaly in at least one of the system's variables. To that end, we use the Tennessee Eastmann (TE) process benchmark (Downs & Vogel, 1993), especially the simulations of Reinartz et al. (2021), pre-processed by Montesuma et al. (2024b).

This benchmark is composed by simulations of a large-scale chemical plant, from which a collection of 34 physical and chemical quantities are measured over time. There are a total of 29 states for this plant: 1 healthy state, and 28 faulty states. Furthermore, the plant can operate under 6 different modes of operation, depending on the specified production requirements. We refer readers to the aforementioned references, as well as our appendix, for detailed descriptions of the variables, faults and modes of operation.

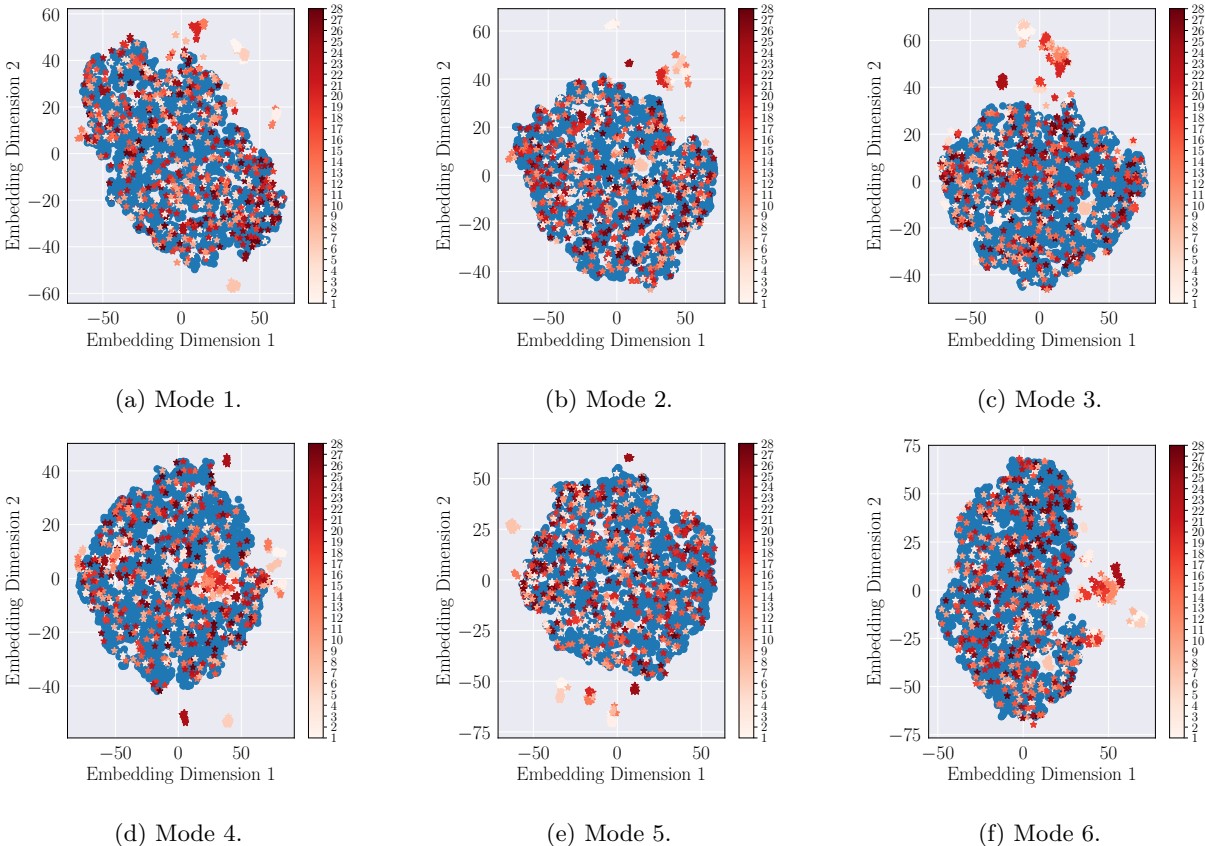

Figure 10: t-SNE embeddings of the Tennessee Eastman process data per mode. In blue, we show the normal samples, whereas we show anomalous samples in shades of reds, corresponding to the actual fault category they correspond. While most anomalous samples do cluster in a region outside the non-faulty cluster, some faulty samples do not.

From the original simulations in Reinartz et al. (2021), we consider 100 simulations for the normal state. These simulations last for 600 hours, with a time-step of 1 hour, and concern 34 sensors measuring different physical and chemical properties. On top of these 100 simulations, we take 1 faulty simulation for each kind of fault, leading to a total of 128 simulations. Based on this set of simulations, we extract windows of 20

hours from the original signals. Each window is then considered a sample for anomaly detection. On each window, we compute the mean, and standard deviation of each variable, leading to vectors $\mu, \sigma \in \mathbb{R}^{34}$. We use the concatenation of these vectors as features for anomaly detection. These steps lead to 6 datasets with 3840 samples, 840 (21.875%) of them being anomalous, one for each mode of operation.

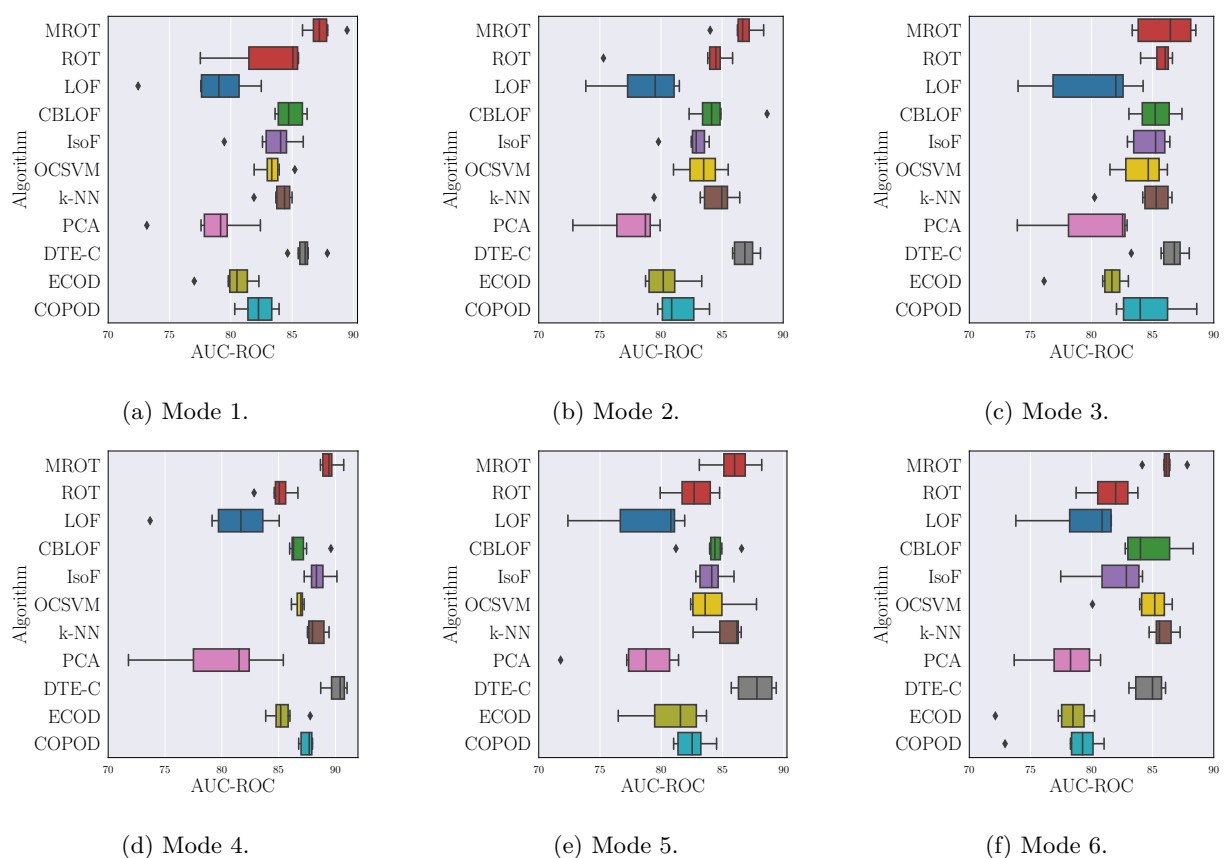

Figure 11: Aggregated anomaly detection on the Tennessee Eastman data per mode of operation. First, MROT outperforms OT with repulsive costs over all modes. Second, MROT and DTE-C have state-of-the-art performance, superior to previously proposed methods.

We start our analysis by embedding the obtained features in $\mathbb{R}^2$ through t-Stochastic Neighbor Embeddings (t-SNE) (Van der Maaten & Hinton, 2008), which is shown in Figure 10. We scatter the embeddings of each mode's data, showing that most faulty samples cluster outside the normal data cluster. However, some faulty samples are close to normal ones. This phenomenon is expected, as the effect of faults evolves over time. As a result, windows taken in early stages of simulation resemble those of normal samples.

Next, we benchmark the AD performance of algorithms, and their capability to generalize to unseen that within the same mode of operation. In this experiment, we downsample the number of anomalous samples per fault category to $\{5, 10, \cdots, 30\}$. This results in a percentage of $\{4.45\%, 8.53\%, 12.28\%, 15.73\%, 18.92\%, 21.87\%\}$ of anomalous samples. In Figure 11, we report our aggregated results over all percentage of anomalies. We refer readers to our appendix for results per percentage.

The main idea of this methodology is evaluating how different algorithms perform, under a variable percentage of faults. Our results are shown in Figure 10 (b). Among the tested methods, MROT has a better performance and remains stable throughout the range of percentage of anomalies.

### 4.3 Ablations

**Hyper-parameter robustness.** Here, we analyze the robustness of our method with respect to the entropic regularization penalty $\epsilon$, and the number of nearest neighbors $k$ in $\mathcal{N}_k$. In our experiments, we evaluated our method on the values $\epsilon \in \{0, 10^{-2}, 10^{-1}, 10^0\}$, where $\epsilon = 0$ implies the use of exact OT, that is, linear programming. For MROT, we use $k \in \{5, 10, 20, \cdots, 50\}$. Note that while $\epsilon$ is related to *transportation plan*, $k$ is linked to the *ground-cost*. We summarize our results in Figure 12, where we present box-plots over the AUC-ROC scores on each dataset in the AdBench benchmark, for each combination of hyper-parameters.

From Figure 12, we note that both MROT is robust to the choice of entropic regularization and number of nearest neighbors. As a general guideline, it is better to limit the number of nearest neighbors, as anomalous examples are likely rare. As a consequence, using a high value for $k$ may lead to the inclusion of normal points in the neighborhood of anomalous ones.

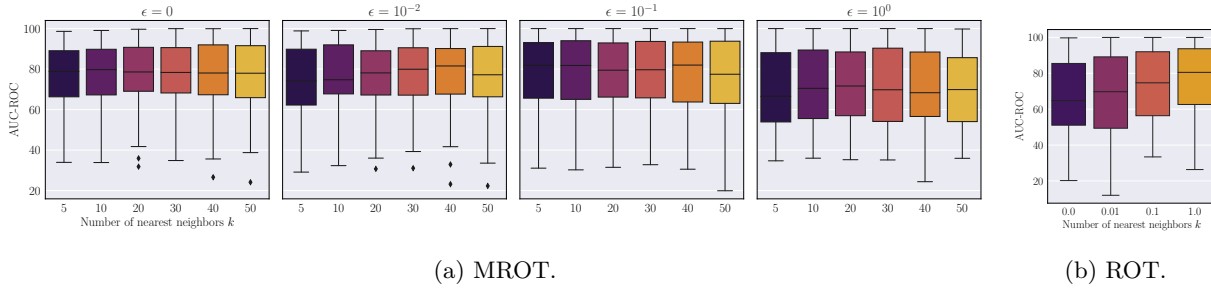

(a) MROT.                                                                                        (b) ROT.

Figure 12: Hyper-parameter sensitivity. In (a), we show the performance of MROT for a variable number of nearest neighbors $k$, and entropic regularization $\epsilon$. Overall, our method is robust to the choices of these hyper-parameters, but using a lower $\epsilon$ is generally better. In contrast, we show in (b) the performance of OT-based AD with the regularized Coulomb cost, for which using a higher entropic penalty $\epsilon$ improves performance.

We give an additional reasoning on the choice of $\epsilon$. From the classical OT (Peyré & Cuturi, 2020, Chapter 4), using $\epsilon > 0$ leads to a nonsparse plan. This means that there are more indices $(i, j)$, for which $\gamma_{ij} > 0$. Likewise, if we let $\epsilon \to +\infty$, then $\gamma$ converges to the trivial coupling $\gamma_{ij} = p_i p_j$. Assuming uniform importance, i.e., $p_i = n^{-1}$, and plugging these results back into equation 10, we have $t_i = n^{-1} \sum_{j=1}^{n} \tilde{c}_k(x_i^{(P)}, x_j^{(P)})$, that is, for a large entropic penalty, we end up simply computing the average cost of each sample. The implication of this analysis is as follows: the MROT plan captures some structure in the relationship between anomalous and normal points, which contributes to a more accurate anomaly score.

**Robustness to regression algorithm.** As we mentioned in our toy example, it is necessary to use a nonlinear regression model for generalizing the anomaly score, as the relationship between $\mathbf{x}_i^{(P)}$ and $\hat{F}_{\mathcal{T}}(t_i)$ is likely nonlinear. We ablate on the choice of regression algorithm, exploring the use of gradient boosting (Friedman, 2002), SVR (Smola & Schölkopf, 2004) and k-nearest neighbors regression. We show a summary of our results in Figure 13. Note that, in general, the performance of boosting and SVR is similar and stable among different choices for $k$ in $\mathcal{N}_k$ in our engineered cost. The performance of $k-$nearest neighbors regression is sub-optimal, especially due to overfitting.

In addition to the pipeline shown in Algorithm 1, it is possible to add a model selection step, e.g., through k-fold cross-validation. The idea is to partition the training dataset $\{\mathbf{x}_i^{(P)}, \hat{F}_{\mathcal{T}}(t_i)\}_{i=1}^n$ into a training and a validation set. Then, among a pool of possible regression models, the one that has the minimum error in the validation set.

## 5 Conclusion

In this paper, we introduced a novel, general purpose anomaly detection algorithm based on optimal transport theory. Our method works under the assumption that the local neighborhood of anomalous samples is likely

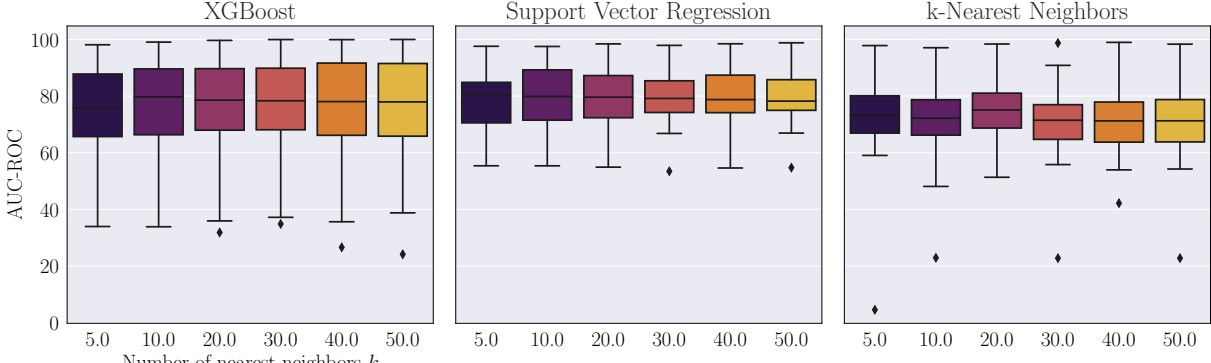

Figure 13: Ablation of MROT performance with respect regression model choice. Overall, gradient boosting and SVR have similar performance, whereas $k-$nearest neighbors has sub-optimal performance due to overfitting.

more irregular than that of normal samples. Based on this idea, we engineer the ground-cost in optimal transport, to encourage samples to send their mass *just outside* an exclusion zone, defined through its $k-$nearest neighbors. While this idea bears some similarity to optimal transport with repulsive costs (Di Marino et al., 2017), the defined ground-cost is not repulsive, and, as we show in our experiments, it leads to better anomaly detectors. We thoroughly experiment on the AdBench (Han et al., 2022) benchmark, and the Tennessee Eastman process (Downs & Vogel, 1993; Reinartz et al., 2021; Montesuma et al., 2024b), showing that our method outperforms previously proposed methods in real-world benchmarks.

**Limitations.** Our methods faces 2 limitations from OT computation. First, we need to store a $n \times m$ matrix, which incurs a $\mathcal{O}(n^2)$ storage complexity. Furthermore, computing $\gamma_\epsilon^\star$ has $\mathcal{O}(n^3 \log n)$ computational complexity. For that reason, in the AdBench experiments, we had to downsample larger datasets to $n = 20,000$ samples. Second, OT is notoriously hard to estimate in higher dimensions. As a result, our method struggles with high-dimensional features, such as those in the NLP datasets of the AdBench benchmark.

In the literature, there are different ways of tackling these limitations, but we leave those for future works. For instance, a workaround to the first limitations is to focus on mini-batch OT (Fatras et al., 2020) or computing OT with neural nets (Makkuva et al., 2020; Korotin et al., 2023). Likewise, the computation of OT in high dimensions can be done, for instance, through modifications of the original OT objective, such as subspace robust OT (Paty & Cuturi, 2019).

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

# A    Additional Details on Experiments

## A.1    Detailed results on AdBench

Table 1: Comparison of AUC-ROC results on ADBench.

| Dataset | IsoF | OCSVM | k-NN | PCA | LOF | CBLOF | ECOD | COPOD | LODA | FeatureBagging | HBOS | OT | MROT | DTE-IG | DTE-NP | DDPM | DTE-C |
|---|---|---|---|---|---|---|---|---|---|---|---|---|---|---|---|---|---|
| cover | 91.48 ± 1.50 | 65.64 ± 0.25 | 88.61 ± 0.39 | 83.12 ± 0.34 | 56.75 ± 1.89 | 92.24 ± 0.17 | 92.02 ± 0.49 | 88.20 ± 0.32 | 92.18 ± 3.85 | 57.14 ± 2.15 | 70.68 ± 1.16 | 92.61 ± 0.28 | 88.01 ± 1.01 | 82.76 ± 1.85 | 81.76 ± 1.75 | 80.76 ± 1.65 | 83.76 ± 1.95 |
| donors | 78.24 ± 1.15 | 70.11 ± 0.14 | 71.91 ± 0.20 | 58.16 ± 0.19 | 62.89 ± 1.35 | 80.77 ± 0.71 | 88.87 ± 0.11 | 81.53 ± 0.27 | 56.62 ± 38.02 | 69.06 ± 1.74 | 74.31 ± 0.63 | 84.85 ± 0.21 | 84.51 ± 0.19 | 82.63 ± 2.08 | 81.63 ± 1.98 | 80.63 ± 1.88 | 83.63 ± 2.18 |
| fault | 57.42 ± 3.35 | 53.42 ± 1.30 | 70.27 ± 2.38 | 48.37 ± 2.12 | 57.88 ± 1.35 | 66.50 ± 2.51 | 45.35 ± 1.77 | 45.49 ± 0.16 | 47.78 ± 2.34 | 59.10 ± 1.16 | 50.62 ± 6.66 | 54.29 ± 3.10 | 56.35 ± 7.46 | 58.22 ± 1.15 | 57.22 ± 1.05 | 56.22 ± 0.95 | 59.22 ± 1.25 |
| fraud | 94.47 ± 1.36 | 94.83 ± 0.38 | 96.57 ± 0.80 | 90.31 ± 0.36 | 54.77 ± 7.30 | 95.39 ± 0.81 | 94.87 ± 0.81 | 94.29 ± 1.42 | 85.55 ± 5.41 | 61.59 ± 7.15 | 94.51 ± 1.22 | 95.08 ± 1.21 | 94.48 ± 1.37 | 94.37 ± 1.57 | 93.37 ± 1.47 | 92.37 ± 1.37 | 95.37 ± 1.67 |
| glass | 80.49 ± 12.70 | 45.44 ± 4.93 | 76.34 ± 9.85 | 55.13 ± 5.13 | 61.76 ± 12.42 | 85.50 ± 4.18 | 75.37 ± 13.70 | 75.95 ± 1.51 | 62.43 ± 12.75 | 65.86 ± 12.47 | 82.02 ± 1.77 | 80.24 ± 10.54 | 78.29 ± 10.58 | 58.04 ± 9.31 | 57.04 ± 9.21 | 56.04 ± 9.11 | 59.04 ± 9.41 |
| Hepatitis | 62.05 ± 13.25 | 48.22 ± 2.80 | 46.67 ± 13.62 | 51.67 ± 6.77 | 33.75 ± 1.07 | 99.61 ± 0.03 | 69.23 ± 6.78 | 99.07 ± 0.26 | 5.95 ± 9.35 | 28.82 ± 1.45 | 99.11 ± 0.16 | 84.62 ± 11.03 | 81.54 ± 14.66 | 99.96 ± 0.36 | 99.76 ± 0.16 | 100.06 ± 0.46 | |
| http | 99.96 ± 0.02 | 99.53 ± 0.01 | 9.41 ± 0.55 | 95.15 ± 0.06 | 54.86 ± 1.33 | 54.77 ± 4.28 | 97.85 ± 0.02 | 42.17 ± 0.12 | 38.23 ± 3.78 | 54.03 ± 1.15 | 57.50 ± 0.58 | 99.46 ± 0.06 | 99.31 ± 0.31 | 51.62 ± 0.79 | 50.62 ± 0.69 | 49.62 ± 0.59 | 52.62 ± 0.89 |
| InternetAds | 70.93 ± 2.22 | 69.77 ± 1.82 | 67.05 ± 2.71 | 58.00 ± 1.40 | 87.81 ± 0.48 | 76.32 ± 1.80 | 67.74 ± 2.69 | 56.02 ± 0.07 | 53.71 ± 4.29 | 88.58 ± 0.51 | 58.85 ± 0.59 | 65.48 ± 3.77 | 68.37 ± 5.97 | 86.73 ± 1.64 | 85.73 ± 1.54 | 84.73 ± 1.44 | 87.73 ± 1.74 |
| Ionosphere | 83.48 ± 1.91 | 82.61 ± 2.58 | 85.36 ± 2.05 | 67.22 ± 3.64 | 67.84 ± 0.38 | 72.53 ± 0.10 | 75.37 ± 6.49 | 68.10 ± 0.04 | 65.50 ± 1.41 | 69.96 ± 0.64 | 70.93 ± 0.55 | 82.90 ± 2.11 | 81.91 ± 3.27 | 78.26 ± 1.68 | 77.26 ± 1.58 | 76.26 ± 1.48 | 79.26 ± 1.78 |
| landsat | 47.52 ± 1.57 | 41.31 ± 1.28 | 61.83 ± 1.88 | 46.74 ± 0.55 | 70.15 ± 1.87 | 79.50 ± 1.84 | 37.35 ± 1.10 | 90.54 ± 0.03 | 86.69 ± 2.45 | 72.61 ± 0.51 | 83.77 ± 0.90 | 43.51 ± 1.58 | 67.72 ± 1.78 | 76.94 ± 2.34 | 75.94 ± 2.24 | 74.94 ± 2.14 | 77.94 ± 2.44 |
| ALOI | 53.27 ± 1.93 | 52.04 ± 0.40 | 60.38 ± 1.59 | 51.99 ± 0.36 | 65.77 ± 1.94 | 84.26 ± 1.11 | 52.10 ± 1.87 | 50.00 ± 0.00 | 56.40 ± 11.07 | 66.44 ± 1.42 | 57.36 ± 0.38 | 54.19 ± 2.11 | 53.69 ± 1.68 | 83.60 ± 1.40 | 82.60 ± 1.30 | 81.60 ± 1.20 | 84.60 ± 1.50 |
| letter | 63.35 ± 6.64 | 50.91 ± 1.92 | 92.54 ± 1.27 | 51.31 ± 0.37 | 58.07 ± 8.89 | 100.00 ± 0.00 | 58.30 ± 2.73 | 94.81 ± 0.48 | 99.30 ± 0.77 | 57.54 ± 10.67 | 100.00 ± 0.00 | 53.11 ± 4.87 | 70.61 ± 5.33 | 99.95 ± 0.23 | 99.95 ± 0.13 | 99.95 ± 0.03 | 99.95 ± 0.33 |
| Lymphography | 100.00 ± 0.00 | 99.37 ± 0.32 | 98.62 ± 3.08 | 94.56 ± 1.30 | 53.82 ± 5.25 | 78.48 ± 0.95 | 98.62 ± 1.89 | 50.00 ± 0.00 | 49.29 ± 9.35 | 53.93 ± 5.59 | 86.81 ± 0.25 | 91.03 ± 13.04 | 99.31 ± 1.54 | 42.15 ± 3.91 | 41.15 ± 3.81 | 40.15 ± 3.71 | 43.15 ± 4.01 |
| magic.gamma | 71.93 ± 0.58 | 67.25 ± 0.35 | 79.89 ± 0.92 | 60.08 ± 0.27 | 53.42 ± 4.61 | 86.38 ± 7.22 | 64.40 ± 0.97 | 90.60 ± 0.43 | 89.51 ± 0.85 | 51.82 ± 5.22 | 92.47 ± 0.38 | 69.54 ± 0.99 | 71.54 ± 0.89 | 72.03 ± 5.63 | 71.03 ± 5.53 | 70.03 ± 5.43 | 73.03 ± 5.73 |
| mammography | 84.95 ± 4.51 | 87.00 ± 0.53 | 86.37 ± 1.72 | 75.40 ± 0.49 | 70.95 ± 1.05 | 67.57 ± 0.98 | 90.33 ± 2.46 | 77.67 ± 0.17 | 45.33 ± 12.81 | 78.77 ± 2.68 | 60.84 ± 2.38 | 88.46 ± 2.27 | 88.24 ± 1.66 | 83.37 ± 1.57 | 82.37 ± 1.47 | 81.37 ± 1.37 | 84.37 ± 1.67 |
| mnist | 80.43 ± 2.20 | 82.28 ± 0.71 | 84.81 ± 0.77 | 67.44 ± 1.85 | 54.98 ± 1.29 | 74.19 ± 3.23 | 74.84 ± 1.00 | 63.34 ± 0.04 | 61.39 ± 2.92 | 54.54 ± 1.26 | 76.18 ± 0.55 | 86.49 ± 1.28 | 88.16 ± 1.05 | 73.53 ± 0.72 | 72.53 ± 0.62 | 71.53 ± 0.52 | 74.53 ± 0.82 |
| musk | 99.97 ± 0.05 | 100.00 ± 0.00 | 90.02 ± 3.10 | 95.91 ± 0.36 | 53.91 ± 7.29 | 99.88 ± 0.01 | 95.26 ± 0.80 | 97.46 ± 0.04 | 98.14 ± 0.49 | 52.55 ± 7.36 | 97.60 ± 0.12 | 100.00 ± 0.00 | 99.97 ± 0.06 | 99.75 ± 0.25 | 99.65 ± 0.15 | 99.55 ± 0.05 | 99.85 ± 0.35 |
| optdigits | 73.76 ± 5.09 | 54.84 ± 2.28 | 39.46 ± 3.03 | 45.23 ± 0.36 | 52.55 ± 1.09 | 62.06 ± 3.96 | 63.04 ± 2.53 | 99.50 ± 0.10 | 38.87 ± 13.16 | 49.29 ± 3.40 | 98.63 ± 0.23 | 55.07 ± 2.92 | 55.21 ± 1.13 | 97.70 ± 2.48 | 97.60 ± 2.38 | 97.50 ± 2.28 | 97.80 ± 2.58 |
| PageBlocks | 90.72 ± 1.82 | 68.81 ± 0.72 | 58.50 ± 2.43 | 72.16 ± 1.98 | 54.97 ± 0.17 | 67.51 ± 3.80 | 90.75 ± 0.89 | 47.12 ± 0.19 | 44.16 ± 1.92 | 53.38 ± 0.29 | 58.84 ± 0.37 | 90.29 ± 0.52 | 90.27 ± 0.72 | 48.09 ± 2.31 | 47.09 ± 2.21 | 46.09 ± 2.11 | 49.09 ± 2.41 |
| pendigits | 94.74 ± 1.12 | 95.81 ± 0.17 | 82.88 ± 1.73 | 78.44 ± 0.66 | 89.92 ± 5.31 | 86.29 ± 4.76 | 92.74 ± 2.46 | 91.17 ± 1.63 | 81.88 ± 3.56 | 79.36 ± 5.06 | 80.91 ± 4.76 | 94.80 ± 1.02 | 88.74 ± 1.09 | 95.96 ± 1.50 | 95.66 ± 1.40 | 95.56 ± 1.30 | 95.86 ± 1.60 |
| Pima | 67.95 ± 1.39 | 66.33 ± 1.73 | 62.42 ± 2.36 | 54.35 ± 0.85 | 51.15 ± 0.65 | 47.11 ± 0.24 | 53.95 ± 5.03 | 48.89 ± 0.29 | 46.57 ± 1.85 | 50.85 ± 0.71 | 47.32 ± 0.19 | 72.81 ± 1.23 | 72.72 ± 0.95 | 48.57 ± 1.37 | 47.57 ± 1.27 | 46.57 ± 1.17 | 49.57 ± 1.47 |
| annthyroid | 81.13 ± 1.56 | 57.20 ± 0.67 | 69.45 ± 2.57 | 60.50 ± 1.09 | 65.69 ± 3.08 | 90.93 ± 1.07 | 79.06 ± 1.70 | 93.91 ± 0.26 | 81.89 ± 12.49 | 70.68 ± 1.69 | 94.84 ± 1.01 | 67.49 ± 0.77 | 57.56 ± 3.28 | 89.06 ± 1.70 | 88.06 ± 1.60 | 87.06 ± 1.50 | 90.06 ± 1.80 |
| satellite | 69.06 ± 1.91 | 66.24 ± 0.37 | 72.76 ± 0.85 | 64.68 ± 1.51 | 48.68 ± 3.40 | 46.34 ± 1.41 | 58.69 ± 1.30 | 26.28 ± 3.94 | 29.44 ± 5.11 | 47.32 ± 5.60 | 31.73 ± 3.46 | 65.36 ± 1.00 | 74.69 ± 1.12 | 58.34 ± 7.51 | 57.34 ± 7.51 | 56.34 ± 7.41 | 59.34 ± 7.71 |
| satimage-2 | 99.25 ± 0.94 | 99.62 ± 0.14 | 99.67 ± 0.25 | 92.63 ± 1.10 | 76.42 ± 2.72 | 89.71 ± 0.72 | 95.73 ± 2.03 | 50.00 ± 0.00 | 51.50 ± 16.40 | 79.03 ± 3.04 | 74.04 ± 0.77 | 99.50 ± 0.47 | 99.85 ± 0.20 | 91.18 ± 0.82 | 90.18 ± 0.72 | 89.18 ± 0.62 | 92.18 ± 0.92 |
| shuttle | 99.72 ± 0.11 | 99.19 ± 0.03 | 79.43 ± 0.48 | 96.46 ± 0.07 | 93.18 ± 1.07 | 88.38 ± 0.50 | 99.29 ± 0.05 | 49.60 ± 0.44 | 70.52 ± 4.82 | 93.29 ± 1.73 | 67.91 ± 0.98 | 99.32 ± 0.30 | 99.06 ± 0.09 | 92.34 ± 2.96 | 91.34 ± 2.86 | 90.34 ± 2.76 | 93.34 ± 3.06 |
| skin | 64.79 ± 1.18 | 54.38 ± 0.33 | 71.22 ± 0.05 | 43.68 ± 0.08 | 32.97 ± 14.63 | 45.27 ± 33.40 | 48.73 ± 0.14 | 86.46 ± 4.66 | 82.17 ± 3.91 | 32.34 ± 5.54 | 90.72 ± 3.31 | 61.75 ± 0.24 | 76.09 ± 0.17 | 39.39 ± 8.57 | 38.39 ± 8.47 | 37.39 ± 8.37 | 40.39 ± 8.67 |
| smtp | 85.78 ± 4.20 | 75.50 ± 3.03 | 92.28 ± 4.06 | 82.41 ± 2.03 | 45.30 ± 2.12 | 46.06 ± 1.23 | 90.46 ± 7.47 | 38.03 ± 0.15 | 46.13 ± 4.68 | 46.45 ± 1.53 | 40.17 ± 0.97 | 89.58 ± 5.74 | 89.82 ± 8.77 | 48.31 ± 0.92 | 47.31 ± 0.82 | 46.31 ± 0.72 | 49.31 ± 1.02 |
| SpamBase | 57.66 ± 1.63 | 51.24 ± 1.00 | 72.70 ± 1.02 | 48.89 ± 0.95 | 44.61 ± 3.25 | 96.08 ± 0.93 | 64.61 ± 0.59 | 99.44 ± 0.16 | 96.97 ± 3.27 | 40.83 ± 2.60 | 98.44 ± 0.30 | 57.07 ± 1.58 | 56.16 ± 1.62 | 78.64 ± 4.42 | 77.64 ± 4.32 | 76.64 ± 4.22 | 79.64 ± 4.52 |
| speech | 54.70 ± 10.76 | 46.74 ± 0.88 | 46.13 ± 8.62 | 50.40 ± 0.98 | 61.44 ± 0.34 | 73.78 ± 0.30 | 45.31 ± 9.31 | 78.28 ± 0.04 | 49.27 ± 8.77 | 59.37 ± 4.19 | 76.81 ± 0.27 | 58.57 ± 10.63 | 61.04 ± 8.76 | 74.38 ± 0.97 | 73.38 ± 0.87 | 72.38 ± 0.77 | 75.38 ± 1.07 |
| Stamps | 90.43 ± 1.49 | 85.28 ± 2.83 | 89.89 ± 3.53 | 63.15 ± 9.14 | 55.12 ± 2.42 | 83.16 ± 1.77 | 88.06 ± 4.22 | 92.08 ± 0.30 | 85.53 ± 7.10 | 57.89 ± 2.77 | 83.94 ± 1.23 | 90.27 ± 1.28 | 89.14 ± 5.02 | 74.33 ± 6.13 | 73.33 ± 6.03 | 72.33 ± 5.93 | 75.33 ± 6.23 |
| thyroid | 97.83 ± 1.10 | 89.35 ± 0.57 | 95.53 ± 1.17 | 87.00 ± 3.16 | 43.21 ± 1.22 | 75.34 ± 1.76 | 97.28 ± 0.54 | 75.70 ± 0.62 | 59.97 ± 11.91 | 51.39 ± 2.70 | 75.41 ± 0.65 | 95.88 ± 0.66 | 95.96 ± 1.65 | 81.58 ± 2.08 | 80.58 ± 1.98 | 79.58 ± 1.88 | 82.58 ± 2.18 |
| vertebral | 36.43 ± 7.36 | 44.36 ± 7.51 | 22.62 ± 11.14 | 43.15 ± 2.60 | 56.20 ± 0.61 | 66.40 ± 0.18 | 14.05 ± 6.31 | 50.00 ± 0.00 | 45.44 ± 13.11 | 53.75 ± 0.31 | 61.09 ± 0.34 | 26.35 ± 1.33 | 23.17 ± 10.55 | 67.87 ± 0.24 | 66.87 ± 0.24 | 65.87 ± 0.14 | 68.87 ± 0.44 |
| backdoor | 73.47 ± 3.27 | 86.32 ± 0.14 | 73.93 ± 0.54 | 84.20 ± 1.07 | 76.66 ± 0.35 | 55.58 ± 0.20 | 84.87 ± 0.58 | 51.53 ± 0.01 | 49.52 ± 1.02 | 79.15 ± 0.56 | 53.11 ± 0.21 | 88.77 ± 0.56 | 88.80 ± 0.84 | 55.22 ± 0.45 | 54.22 ± 0.35 | 53.22 ± 0.25 | 56.22 ± 0.55 |
| vowels | 77.30 ± 2.40 | 77.17 ± 1.34 | 93.79 ± 1.71 | 59.48 ± 2.51 | 52.67 ± 2.09 | 56.09 ± 2.47 | 56.87 ± 4.12 | 66.42 ± 3.42 | 70.80 ± 13.23 | 53.79 ± 1.73 | 59.50 ± 1.09 | 63.52 ± 3.49 | 93.55 ± 3.05 | 59.86 ± 4.47 | 58.86 ± 4.37 | 57.86 ± 4.27 | 60.86 ± 4.57 |
| Waveform | 71.16 ± 3.99 | 66.83 ± 2.93 | 78.04 ± 1.81 | 52.53 ± 0.90 | 46.77 ± 8.48 | 63.47 ± 14.61 | 61.45 ± 4.70 | 80.74 ± 0.93 | 55.74 ± 15.73 | 46.94 ± 11.14 | 76.77 ± 1.57 | 74.26 ± 3.29 | 71.09 ± 3.44 | 48.12 ± 8.23 | 47.12 ± 8.13 | 46.12 ± 8.03 | 49.12 ± 8.33 |
| WBC | 99.77 ± 0.52 | 99.36 ± 0.29 | 100.00 ± 0.00 | 95.44 ± 1.53 | 58.73 ± 1.53 | 61.55 ± 0.12 | 100.00 ± 0.00 | 67.61 ± 0.07 | 54.06 ± 5.45 | 49.42 ± 4.74 | 69.56 ± 0.04 | 98.84 ± 0.82 | 99.07 ± 0.52 | 63.37 ± 0.23 | 62.37 ± 0.13 | 61.37 ± 0.03 | 64.37 ± 0.33 |
| WDBC | 97.64 ± 3.17 | 99.83 ± 0.02 | 99.86 ± 0.31 | 95.70 ± 1.57 | 86.38 ± 2.12 | 89.19 ± 1.14 | 96.53 ± 2.35 | 78.27 ± 1.27 | 78.84 ± 4.28 | 87.64 ± 1.63 | 54.44 ± 3.14 | 98.61 ± 1.47 | 98.75 ± 0.91 | 77.75 ± 2.61 | 76.75 ± 2.51 | 75.75 ± 2.41 | 78.75 ± 2.71 |
| Wilt | 44.25 ± 3.33 | 39.65 ± 2.14 | 68.71 ± 3.11 | 45.61 ± 0.49 | 63.63 ± 18.01 | 99.35 ± 0.74 | 38.47 ± 4.20 | 99.60 ± 0.09 | 90.04 ± 11.06 | 52.34 ± 18.31 | 99.49 ± 0.18 | 33.93 ± 2.51 | 32.91 ± 2.10 | 97.77 ± 4.22 | 96.77 ± 4.12 | 95.77 ± 4.02 | 98.77 ± 4.32 |
| wine | 82.08 ± 9.50 | 99.92 ± 0.12 | 100.00 ± 0.00 | 66.04 ± 11.09 | 70.33 ± 1.55 | 89.30 ± 2.14 | 73.75 ± 20.60 | 87.48 ± 0.21 | 71.24 ± 10.47 | 75.83 ± 1.79 | 77.89 ± 2.07 | 96.67 ± 3.78 | 90.00 ± 6.49 | 84.00 ± 1.11 | 83.00 ± 1.01 | 82.00 ± 0.91 | 85.00 ± 1.21 |
| WPBC | 49.53 ± 16.13 | 47.74 ± 2.20 | 51.33 ± 3.43 | 46.44 ± 3.00 | 56.34 ± 3.89 | 65.49 ± 1.65 | 47.17 ± 2.92 | 66.20 ± 1.21 | 59.52 ± 7.09 | 57.31 ± 3.88 | 70.38 ± 1.21 | 45.81 ± 6.76 | 58.06 ± 11.96 | 55.74 ± 5.49 | 54.74 ± 5.39 | 53.74 ± 5.29 | 56.74 ± 5.59 |
| yeast | 42.06 ± 2.20 | 42.67 ± 1.39 | 37.02 ± 3.11 | 49.48 ± 0.57 | 45.29 ± 0.42 | 54.13 ± 0.68 | 43.23 ± 3.54 | 68.79 ± 0.08 | 47.98 ± 10.23 | 42.41 ± 0.84 | 66.41 ± 1.18 | 38.50 ± 3.34 | 41.68 ± 3.77 | 53.01 ± 1.71 | 52.01 ± 1.61 | 51.01 ± 1.51 | 54.01 ± 1.81 |
| breastw | 98.72 ± 0.60 | 93.54 ± 1.39 | 99.09 ± 0.57 | 66.23 ± 2.81 | 51.18 ± 9.24 | 66.04 ± 2.92 | 99.06 ± 0.50 | 92.92 ± 0.83 | 83.10 ± 9.15 | 50.56 ± 7.61 | 90.40 ± 1.00 | 99.42 ± 0.27 | 99.17 ± 0.58 | 57.55 ± 13.03 | 56.55 ± 12.93 | 55.55 ± 12.83 | 58.55 ± 13.13 |
| campaign | 71.14 ± 0.89 | 66.42 ± 0.21 | 72.19 ± 0.86 | 62.54 ± 0.83 | 69.26 ± 1.81 | 70.14 ± 0.99 | 77.42 ± 0.54 | 73.89 ± 0.40 | 59.40 ± 6.16 | 71.52 ± 2.12 | 69.38 ± 0.64 | 73.97 ± 0.33 | 75.07 ± 1.24 | 62.72 ± 3.51 | 61.72 ± 3.41 | 60.72 ± 3.31 | 63.72 ± 3.71 |
| cardio | 93.06 ± 1.79 | 93.04 ± 0.64 | 84.28 ± 2.89 | 81.24 ± 2.33 | 60.66 ± 8.68 | 97.71 ± 1.10 | 93.53 ± 0.69 | 99.40 ± 0.10 | 99.17 ± 0.13 | 38.83 ± 10.40 | 98.70 ± 0.16 | 95.41 ± 0.93 | 82.53 ± 2.40 | 96.80 ± 1.36 | 95.80 ± 1.26 | 94.80 ± 1.16 | 97.80 ± 1.46 |
| Cardiotocography | 72.66 ± 2.96 | 78.33 ± 1.17 | 62.76 ± 2.08 | 59.71 ± 1.64 | 84.87 ± 9.38 | 98.98 ± 0.42 | 80.09 ± 2.85 | 99.29 ± 0.14 | 98.04 ± 0.77 | 86.65 ± 8.37 | 98.94 ± 0.24 | 66.48 ± 1.33 | 49.87 ± 4.02 | 98.54 ± 1.13 | 97.54 ± 1.03 | 96.54 ± 0.93 | 99.54 ± 1.23 |
| celeba | 69.95 ± 1.93 | 68.50 ± 0.27 | 60.34 ± 0.86 | 70.20 ± 0.36 | 67.81 ± 1.65 | 39.59 ± 1.76 | 75.60 ± 0.68 | 34.49 ± 0.27 | 31.31 ± 9.90 | 66.59 ± 7.27 | 34.81 ± 2.83 | 80.54 ± 1.43 | 78.18 ± 0.55 | 67.90 ± 4.12 | 66.90 ± 4.02 | 65.90 ± 3.92 | 68.90 ± 4.22 |
| census | 61.88 ± 2.97 | 53.35 ± 0.15 | 64.49 ± 0.24 | 50.34 ± 0.10 | 44.66 ± 3.87 | 48.68 ± 3.14 | 66.00 ± 0.31 | 51.87 ± 3.09 | 50.07 ± 3.44 | 43.53 ± 3.14 | 54.81 ± 2.85 | 65.92 ± 0.39 | 65.74 ± 0.32 | 52.22 ± 3.37 | 51.22 ± 3.37 | 50.22 ± 3.27 | 53.22 ± 3.57 |
| Average | 76.35 | 71.86 | 73.56 | 66.36 | 60.13 | 74.76 | 74.12 | 72.56 | 63.90 | 59.78 | 73.81 | 75.97 | 77.36 | 73.21 | 72.31 | 71.41 | 74.12 |
| 20news | 55.00 ± 1.81 | 58.29 ± 2.78 | 56.65 ± 1.23 | 54.48 ± 0.69 | 60.98 ± 1.65 | 56.36 ± 1.44 | 54.42 ± 0.42 | 53.26 ± 0.59 | 53.85 ± 4.56 | 60.97 ± 1.63 | 53.69 ± 0.62 | 55.65 ± 1.52 | 52.38 ± 11.98 | 52.72 ± 5.59 | 56.98 ± 1.59 | 54.74 ± 0.95 | 57.87 ± 3.45 |
| agnews | 58.43 ± 1.32 | 66.50 ± 0.50 | 64.65 ± 0.10 | 56.61 ± 0.08 | 71.36 ± 0.64 | 61.91 ± 0.31 | 55.21 ± 0.04 | 55.10 ± 0.04 | 56.81 ± 3.57 | 71.50 ± 0.62 | 55.40 ± 0.08 | 57.20 ± 1.35 | 62.67 ± 6.69 | 54.45 ± 5.22 | 63.22 ± 0.31 | 57.06 ± 0.10 | 62.66 ± 3.75 |
| amazon | 55.76 ± 0.65 | 56.47 ± 0.10 | 60.27 ± 0.04 | 54.95 ± 0.10 | 57.09 ± 0.56 | 57.92 ± 0.24 | 54.10 ± 0.05 | 57.05 ± 0.06 | 52.63 ± 3.04 | 57.18 ± 0.49 | 56.30 ± 0.08 | 55.63 ± 2.08 | 56.42 ± 1.95 | 53.45 ± 1.61 | 60.30 ± 0.40 | 55.13 ± 0.09 | 55.64 ± 2.54 |
| imdb | 48.93 ± 0.66 | 50.36 ± 0.21 | 49.44 ± 0.14 | 47.82 ± 0.06 | 50.02 ± 0.55 | 49.56 ± 0.21 | 47.05 ± 0.03 | 51.20 ± 0.04 | 46.60 ± 2.52 | 49.89 ± 0.54 | 49.86 ± 0.08 | 48.64 ± 1.96 | 49.44 ± 2.48 | 46.61 ± 1.27 | 49.49 ± 0.30 | 47.79 ± 0.11 | 48.44 ± 2.75 |
| yelp | 60.15 ± 0.34 | 65.49 ± 0.64 | 67.01 ± 0.17 | 59.19 ± 0.10 | 66.11 ± 0.48 | 63.53 ± 0.34 | 57.78 ± 0.04 | 60.52 ± 0.15 | 58.10 ± 2.89 | 66.10 ± 0.42 | 59.97 ± 0.11 | 59.06 ± 3.66 | 63.92 ± 1.84 | 51.37 ± 3.24 | 67.08 ± 0.37 | 59.38 ± 0.12 | 60.16 ± 3.22 |
| Average | 55.65 | 59.42 | 59.60 | 54.61 | 61.11 | 57.86 | 53.72 | 55.43 | 53.60 | 61.13 | 55.04 | 55.24 | 56.98 | 52.12 | 59.81 | 54.82 | 56.95 |

Table 2: Comparison of AUC-PR results on ADBench.

| Dataset | IsoF | OCSVM | k-NN | PCA | LOF | CBLOF | ECOD | COPOD | LODA | FeatureBagging | HBOS | OT | MROT | DTE-IG | DTE-NP | DDPM | DTE-C |
|---|---|---|---|---|---|---|---|---|---|---|---|---|---|---|---|---|---|
| cover | 5.18 ± 1.49 | 9.91 ± 0.36 | 5.44 ± 0.56 | 7.53 ± 0.43 | 1.87 ± 0.14 | 6.99 ± 0.28 | 11.25 ± 1.07 | 6.79 ± 0.54 | 9.28 ± 3.80 | 1.90 ± 0.46 | 2.63 ± 0.32 | 5.75 ± 0.95 | 2.49 ± 0.69 | 4.78 ± 0.58 | 4.55 ± 0.85 | 2.10 ± 0.52 | |
| donors | 12.40 ± 0.93 | 13.94 ± 0.30 | 18.21 ± 0.17 | 16.61 ± 0.62 | 10.86 ± 0.23 | 14.77 ± 0.42 | 26.47 ± 0.61 | 20.94 ± 0.53 | 25.47 ± 32.62 | 12.04 ± 0.75 | 13.47 ± 1.11 | 17.81 ± 0.34 | 17.77 ± 0.46 | 16.35 ± 6.18 | 18.83 ± 0.17 | 14.33 ± 0.84 | 13.95 ± 3.79 |
| fault | 39.45 ± 0.61 | 40.08 ± 0.34 | 52.21 ± 0.73 | 33.16 ± 0.61 | 38.77 ± 1.09 | 47.30 ± 3.10 | 32.54 ± 0.17 | 31.26 ± 0.16 | 33.65 ± 2.46 | 39.57 ± 1.22 | 35.97 ± 6.41 | 38.62 ± 2.92 | 41.86 ± 5.36 | 41.70 ± 3.59 | 53.23 ± 0.39 | 39.20 ± 0.65 | 42.18 ± 2.19 |
| fraud | 14.49 ± 5.34 | 10.98 ± 1.28 | 16.86 ± 5.39 | 14.91 ± 3.13 | 0.26 ± 0.05 | 14.53 ± 3.21 | 21.54 ± 4.92 | 25.17 ± 5.67 | 14.62 ± 5.25 | 0.34 ± 0.07 | 20.88 ± 5.54 | 20.42 ± 3.60 | 19.92 ± 6.02 | 18.81 ± 8.21 | 13.68 ± 4.10 | 14.58 ± 3.63 | 64.75 ± 5.97 |
| glass | 14.41 ± 8.02 | 12.98 ± 4.31 | 16.74 ± 2.54 | 11.18 ± 3.06 | 14.42 ± 6.70 | 14.36 ± 3.18 | 18.33 ± 6.17 | 11.05 ± 2.35 | 8.99 ± 3.19 | 15.09 ± 5.90 | 16.07 ± 4.53 | 25.51 ± 13.16 | 18.36 ± 10.36 | 13.54 ± 7.27 | 20.57 ± 6.59 | 7.29 ± 1.13 | 16.82 ± 4.63 |
| hepatitis | 24.31 ± 2.23 | 27.70 ± 3.25 | 25.17 ± 5.46 | 33.91 ± 5.86 | 21.39 ± 6.04 | 30.36 ± 15.57 | 29.47 ± 2.74 | 38.88 ± 3.25 | 27.47 ± 8.72 | 22.49 ± 8.34 | 32.80 ± 4.31 | 16.75 ± 2.15 | 53.62 ± 2.63 | 21.49 ± 8.14 | 23.23 ± 2.75 | 16.49 ± 2.95 | 25.73 ± 7.50 |
| http | 88.63 ± 15.26 | 35.59 ± 2.55 | 0.98 ± 0.69 | 49.99 ± 2.19 | 4.95 ± 2.10 | 46.43 ± 3.33 | 14.47 ± 0.73 | 28.02 ± 4.37 | 0.41 ± 0.07 | 4.69 ± 1.83 | 30.19 ± 3.04 | 19.28 ± 0.54 | 28.89 ± 1.60 | 29.53 ± 19.62 | 2.41 ± 0.99 | 64.22 ± 20.75 | 44.03 ± 16.11 |
| internetads | 48.62 ± 4.28 | 29.09 ± 0.14 | 29.64 ± 0.09 | 27.56 ± 0.76 | 23.20 ± 1.20 | 29.65 ± 0.08 | 50.54 ± 0.20 | 50.47 ± 0.20 | 24.18 ± 3.77 | 18.19 ± 1.90 | 52.27 ± 0.30 | 7.12 ± 0.91 | 12.17 ± 3.04 | 17.53 ± 2.43 | 29.11 ± 0.36 | 29.46 ± 0.05 | 30.18 ± 2.54 |
| ionosphere | 77.92 ± 2.90 | 82.91 ± 0.84 | 91.09 ± 0.79 | 72.08 ± 2.42 | 80.67 ± 4.01 | 88.10 ± 2.92 | 63.34 ± 2.30 | 66.28 ± 3.20 | 74.07 ± 1.71 | 82.05 ± 3.00 | 35.26 ± 2.03 | 62.08 ± 1.12 | 66.43 ± 0.98 | 96.03 ± 1.63 | 92.04 ± 1.18 | 63.29 ± 3.49 | 97.96 ± 2.24 |
| landsat | 19.37 ± 0.64 | 17.50 ± 0.05 | 25.75 ± 0.22 | 16.33 ± 0.13 | 24.99 ± 0.55 | 21.23 ± 1.78 | 16.37 ± 0.05 | 17.60 ± 0.05 | 20.59 ± 0.31 | 24.63 ± 0.49 | 23.07 ± 0.25 | 24.53 ± 7.77 | 19.14 ± 4.58 | 20.27 ± 3.23 | 25.45 ± 0.27 | 19.99 ± 0.35 | 22.34 ± 0.89 |
| aloi | 3.39 ± 0.03 | 3.92 ± 0.14 | 4.76 ± 0.02 | 3.72 ± 0.03 | 9.69 ± 0.28 | 3.74 ± 0.07 | 3.29 ± 0.00 | 3.13 ± 0.00 | 3.29 ± 0.20 | 10.36 ± 0.45 | 3.38 ± 0.03 | 37.03 ± 2.19 | 41.98 ± 2.05 | 3.55 ± 0.55 | 5.55 ± 0.02 | 3.28 ± 0.07 | |
| letter | 8.59 ± 0.16 | 11.27 ± 0.27 | 20.31 ± 0.72 | 7.62 ± 0.12 | 43.32 ± 2.85 | 16.64 ± 1.01 | 7.71 ± 0.07 | 6.84 ± 0.03 | 8.26 ± 1.17 | 44.53 ± 3.22 | 7.79 ± 0.21 | 100.00 ± 0.00 | 99.54 ± 1.04 | 18.09 ± 2.29 | 25.53 ± 1.41 | 36.69 ± 1.64 | 25.65 ± 1.60 |
| lymphography | 97.22 ± 1.71 | 88.48 ± 6.11 | 89.44 ± 6.58 | 93.51 ± 4.78 | 13.52 ± 9.57 | 91.49 ± 6.57 | 89.39 ± 2.04 | 90.69 ± 2.49 | 49.05 ± 38.67 | 74.98 ± 3.41 | 9.00 ± 7.08 | 91.91 ± 3.02 | 3.01 ± 0.08 | 38.80 ± 19.67 | 80.51 ± 9.21 | 73.10 ± 20.13 | 18.53 ± 14.88 |
| magic.gamma | 63.77 ± 0.37 | 62.51 ± 0.10 | 72.35 ± 0.14 | 58.88 ± 0.09 | 51.98 ± 0.44 | 66.61 ± 0.05 | 53.34 ± 0.05 | 58.80 ± 0.04 | 57.87 ± 1.31 | 53.87 ± 0.79 | 61.74 ± 0.15 | 33.44 ± 4.76 | 15.03 ± 4.55 | 65.74 ± 2.99 | 73.98 ± 0.13 | 65.14 ± 1.91 | 66.40 ± 0.97 |
| mammography | 21.78 ± 3.74 | 18.69 ± 0.74 | 18.06 ± 0.92 | 20.44 ± 1.39 | 8.48 ± 0.72 | 13.95 ± 2.79 | 43.54 ± 0.39 | 43.02 ± 0.31 | 31.76 ± 4.58 | 7.01 ± 0.99 | 13.24 ± 1.35 | 19.44 ± 2.11 | 19.48 ± 4.25 | 8.70 ± 2.52 | 17.45 ± 0.99 | 9.89 ± 2.26 | 17.02 ± 1.42 |
| mnist | 29.03 ± 4.81 | 38.54 ± 0.33 | 48.07 ± 0.50 | 38.14 ± 0.94 | 23.34 ± 1.51 | 38.61 ± 1.75 | 9.21 ± 0.00 | 9.16 ± 0.00 | 16.97 ± 6.79 | 24.11 ± 1.05 | 10.91 ± 0.12 | 67.00 ± 1.18 | 50.18 ± 1.53 | 37.54 ± 0.03 | 39.99 ± 0.75 | 37.38 ± 1.09 | 36.76 ± 2.05 |
| musk | 94.47 ± 9.05 | 100.00 ± 0.00 | 70.81 ± 10.35 | 99.95 ± 0.02 | 11.77 ± 5.21 | 100.00 ± 0.00 | 47.47 ± 1.53 | 36.91 ± 4.05 | 84.15 ± 17.56 | 13.95 ± 7.85 | 99.87 ± 0.08 | 93.40 ± 4.12 | 96.91 ± 3.39 | 13.68 ± 4.74 | 43.36 ± 3.14 | 98.38 ± 1.16 | 55.30 ± 21.58 |
| optdigits | 4.61 ± 0.81 | 2.65 ± 0.08 | 2.18 ± 0.09 | 2.70 ± 0.03 | 3.53 ± 0.69 | 5.26 ± 2.66 | 2.88 ± 0.00 | 2.88 ± 0.00 | 2.90 ± 0.95 | 3.62 ± 0.78 | 19.18 ± 1.06 | 96.77 ± 0.64 | 95.43 ± 0.80 | 2.82 ± 0.35 | 2.14 ± 0.09 | 2.24 ± 0.14 | 2.75 ± 0.38 |
| pageblocks | 46.01 ± 4.72 | 22.57 ± 1.29 | 9.95 ± 2.61 | 21.86 ± 0.32 | 4.01 ± 0.53 | 19.17 ± 10.22 | 26.96 ± 0.57 | 17.71 ± 1.05 | 18.56 ± 5.48 | 4.83 ± 0.78 | 25.00 ± 0.90 | 7.20 ± 2.74 | 9.31 ± 2.00 | 7.20 ± 2.74 | 8.87 ± 1.47 | 5.61 ± 0.64 | 4.36 ± 4.56 |
| pima | 50.96 ± 4.11 | 47.74 ± 2.79 | 52.99 ± 3.09 | 49.19 ± 4.08 | 40.63 ± 2.05 | 48.38 ± 3.73 | 48.38 ± 2.46 | 52.63 ± 2.38 | 40.39 ± 5.19 | 41.22 ± 3.23 | 57.73 ± 2.72 | 3.62 ± 1.94 | 3.91 ± 1.22 | 43.74 ± 3.29 | 53.83 ± 2.87 | 40.02 ± 2.72 | 44.68 ± 2.50 |
| annthyroid | 31.23 ± 3.56 | 18.75 ± 0.28 | 22.41 ± 0.47 | 19.55 ± 1.07 | 16.33 ± 0.53 | 16.94 ± 0.78 | 27.21 ± 0.44 | 17.43 ± 0.19 | 9.80 ± 2.71 | 20.55 ± 4.61 | 22.79 ± 0.86 | 30.35 ± 3.41 | 41.37 ± 5.07 | 38.03 ± 6.20 | 22.82 ± 0.34 | 29.74 ± 2.36 | 67.01 ± 0.84 |
| satellite | 64.88 ± 1.51 | 65.44 ± 0.16 | 58.16 ± 0.35 | 60.61 ± 0.17 | 38.10 ± 0.70 | 65.64 ± 0.27 | 52.62 ± 0.10 | 57.04 ± 0.08 | 61.27 ± 4.30 | 37.77 ± 0.72 | 68.78 ± 0.47 | 9.70 ± 0.70 | 9.39 ± 1.14 | 37.96 ± 2.66 | 66.16 ± 0.76 | 52.91 ± 3.46 | |
| satimage-2 | 91.75 ± 0.85 | 96.53 ± 0.02 | 68.98 ± 15.78 | 87.19 ± 0.10 | 4.08 ± 2.50 | 97.21 ± 0.03 | 66.62 ± 1.58 | 79.70 ± 0.94 | 85.74 ± 7.48 | 4.23 ± 2.71 | 76.00 ± 1.14 | 51.57 ± 2.15 | 53.07 ± 2.37 | 9.52 ± 5.66 | 50.73 ± 8.98 | 78.25 ± 5.90 | 13.84 ± 3.38 |
| shuttle | 97.62 ± 0.41 | 90.72 ± 0.06 | 19.31 ± 0.46 | 91.33 ± 0.15 | 50.50 ± 0.14 | 90.50 ± 0.14 | 96.22 ± 0.17 | 16.83 ± 19.06 | 24.88 ± 6.76 | 20.68 ± 0.24 | 23.20 ± 0.19 | 84.00 ± 15.88 | 66.86 ± 6.23 | 31.57 ± 3.11 | 98.21 ± 0.30 | 66.16 ± 0.76 | 13.84 ± 3.38 |
| skin | 25.36 ± 0.43 | 22.01 ± 0.19 | 39.00 ± 0.18 | 17.24 ± 0.15 | 22.10 ± 0.19 | 28.86 ± 3.15 | 18.27 ± 0.10 | 17.86 ± 0.09 | 18.03 ± 0.48 | 20.68 ± 0.24 | 23.20 ± 0.19 | 84.00 ± 15.88 | 66.86 ± 6.23 | 31.57 ± 3.14 | 28.99 ± 0.20 | 17.53 ± 0.88 | 30.24 ± 1.74 |
| smtp | 0.53 ± 0.08 | 38.25 ± 8.36 | 41.54 ± 5.59 | 38.24 ± 5.87 | 2.23 ± 1.39 | 40.32 ± 5.33 | 58.85 ± 4.72 | 0.50 ± 0.05 | 31.21 ± 10.41 | 0.13 ± 0.02 | 0.50 ± 0.05 | 31.40 ± 2.19 | 31.59 ± 2.30 | 1.16 ± 2.20 | 41.07 ± 5.45 | 50.23 ± 9.75 | 42.15 ± 3.73 |
| spambase | 48.75 ± 1.64 | 40.21 ± 0.07 | 41.53 ± 0.17 | 40.93 ± 0.51 | 35.95 ± 0.33 | 40.23 ± 0.63 | 51.82 ± 0.17 | 54.71 ± 0.38 | 43.63 ± 5.96 | 34.39 ± 0.60 | 51.77 ± 1.22 | 39.98 ± 3.19 | 40.69 ± 0.22 | 38.37 ± 0.71 | 40.84 ± 1.52 | | |
| speech | 2.05 ± 0.34 | 1.85 ± 0.03 | 1.85 ± 0.02 | 1.84 ± 0.00 | 2.16 ± 0.15 | 1.87 ± 0.02 | 1.96 ± 0.01 | 1.88 ± 0.07 | 1.61 ± 0.20 | 2.18 ± 0.15 | 2.29 ± 0.14 | 30.27 ± 0.43 | 31.64 ± 2.19 | 1.90 ± 0.21 | 2.04 ± 0.43 | 2.00 ± 0.33 | |
| stamps | 34.72 ± 4.50 | 31.76 ± 4.47 | 31.69 ± 3.92 | 36.40 ± 6.13 | 5.27 ± 4.40 | 30.25 ± 6.21 | 32.35 ± 3.22 | 39.78 ± 4.75 | 27.97 ± 8.26 | 14.26 ± 4.10 | 33.18 ± 3.90 | 71.58 ± 6.37 | 46.30 ± 6.00 | 23.48 ± 11.01 | 27.25 ± 4.34 | 14.26 ± 4.14 | 22.63 ± 4.49 |
| thyroid | 54.32 ± 4.50 | 32.89 ± 2.07 | 39.22 ± 2.16 | 35.57 ± 3.87 | 7.73 ± 2.47 | 27.17 ± 0.59 | 57.18 ± 2.90 | 17.94 ± 0.90 | 18.98 ± 7.40 | 6.93 ± 2.88 | 12.37 ± 3.08 | 9.12 ± 1.03 | 8.43 ± 0.03 | 4.13 ± 3.48 | 13.31 ± 5.54 | 9.82 ± 1.03 | 70.49 ± 4.14 |
| vertebral | 9.68 ± 1.00 | 10.68 ± 1.32 | 9.53 ± 1.18 | 9.93 ± 0.89 | 12.35 ± 3.05 | 12.34 ± 0.98 | 10.97 ± 0.72 | 8.50 ± 1.20 | 8.88 ± 1.14 | 12.37 ± 3.08 | 9.12 ± 1.03 | 8.42 ± 0.10 | 8.43 ± 0.11 | 13.31 ± 5.54 | 9.82 ± 1.03 | 11.92 ± 1.70 | |
| backdoor | 4.54 ± 0.72 | 53.38 ± 1.03 | 47.92 ± 1.45 | 53.14 ± 1.28 | 35.80 ± 2.43 | 54.65 ± 1.42 | 2.48 ± 0.05 | 2.48 ± 0.05 | 10.08 ± 7.77 | 21.68 ± 6.06 | 5.15 ± 0.09 | 66.40 ± 20.45 | 58.08 ± 27.93 | 43.84 ± 3.25 | 52.01 ± 0.96 | 48.07 ± 1.28 | |
| vowels | 16.23 ± 6.18 | 19.58 ± 1.16 | 44.32 ± 0.55 | 6.87 ± 0.26 | 32.58 ± 5.97 | 16.61 ± 1.03 | 8.28 ± 0.54 | 3.43 ± 0.05 | 12.72 ± 3.85 | 31.42 ± 8.14 | 7.83 ± 0.89 | 46.75 ± 2.79 | 49.03 ± 6.90 | 16.57 ± 4.43 | 50.44 ± 3.18 | 31.06 ± 4.37 | 41.70 ± 12.32 |
| waveform | 5.63 ± 0.92 | 5.23 ± 0.11 | 13.28 ± 0.76 | 4.41 ± 0.02 | 7.09 ± 0.90 | 12.23 ± 1.76 | 4.04 ± 0.03 | 5.69 ± 0.14 | 4.02 ± 0.78 | 7.84 ± 1.43 | 4.83 ± 0.11 | 74.10 ± 4.67 | 3.73 ± 0.96 | 10.93 ± 1.18 | 4.98 ± 0.61 | 4.28 ± 0.59 | |
| wbc | 94.84 ± 2.02 | 81.27 ± 11.50 | 74.27 ± 6.66 | 91.33 ± 4.96 | 72.62 ± 1.47 | 69.07 ± 13.79 | 89.19 ± 2.42 | 88.53 ± 2.34 | 89.76 ± 2.29 | 72.83 ± 6.35 | 3.42 ± 0.22 | 3.46 ± 3.91 | 74.01 ± 4.67 | 4.78 ± 0.40 | 71.72 ± 17.79 | 72.17 ± 13.60 | 75.78 ± 9.25 |
| wdbc | 70.18 ± 4.66 | 53.89 ± 7.79 | 52.13 ± 4.05 | 61.28 ± 3.38 | 12.60 ± 7.89 | 68.81 ± 9.09 | 49.27 ± 4.01 | 76.04 ± 3.54 | 52.69 ± 13.22 | 15.45 ± 9.61 | 58.64 ± 9.40 | 57.14 ± 12.33 | 74.89 ± 4.49 | 25.66 ± 4.20 | 14.04 ± 6.34 | 28.37 ± 10.80 | 15.65 ± 7.72 |
| wilt | 4.40 ± 0.25 | 3.54 ± 0.01 | 4.92 ± 0.07 | 3.22 ± 0.01 | 8.31 ± 0.34 | 4.11 ± 0.12 | 4.17 ± 0.00 | 3.70 ± 0.01 | 3.60 ± 0.48 | 8.05 ± 2.16 | 3.94 ± 0.15 | 3.12 ± 0.18 | 3.36 ± 0.65 | 5.35 ± 0.07 | 7.62 ± 0.85 | 16.29 ± 1.47 | |
| wine | 20.69 ± 4.89 | 13.48 ± 2.11 | 8.05 ± 0.89 | 26.39 ± 5.02 | 6.42 ± 1.66 | 17.04 ± 22.72 | 19.45 ± 3.20 | 36.39 ± 6.24 | 24.99 ± 9.90 | 6.06 ± 0.50 | 41.21 ± 10.01 | 55.00 ± 3.12 | 51.96 ± 2.29 | 6.39 ± 1.93 | 7.37 ± 1.21 | 7.45 ± 2.14 | 10.27 ± 3.41 |
| wpbc | 23.73 ± 1.92 | 22.15 ± 1.31 | 23.44 ± 1.40 | 22.86 ± 1.58 | 20.98 ± 1.73 | 22.74 ± 1.64 | 21.66 ± 1.22 | 23.37 ± 1.68 | 21.53 ± 1.49 | 20.57 ± 1.44 | 24.10 ± 1.66 | 42.16 ± 1.82 | 42.49 ± 1.99 | 21.55 ± 2.60 | 22.37 ± 2.01 | 23.30 ± 3.11 | 23.14 ± 3.11 |
| yeast | 30.39 ± 0.49 | 30.33 ± 0.38 | 29.36 ± 0.48 | 30.17 ± 0.20 | 31.51 ± 0.78 | 31.39 ± 0.56 | 33.19 ± 0.18 | 30.79 ± 0.15 | 33.01 ± 2.79 | 32.55 ± 0.99 | 32.79 ± 0.50 | 37.40 ± 7.46 | 34.67 ± 9.43 | 30.64 ± 1.82 | 29.45 ± 0.66 | 32.04 ± 0.49 | 30.61 ± 1.70 |
| breastw | 95.64 ± 1.34 | 89.69 ± 1.55 | 93.20 ± 1.85 | 57.65 ± 2.09 | 38.98 ± 3.32 | 96.24 ± 0.39 | 98.87 ± 0.33 | 95.50 ± 3.15 | 86.27 ± 5.17 | 35.35 ± 4.47 | 14.51 ± 2.47 | 35.21 ± 0.82 | 85.00 ± 9.13 | 86.67 ± 7.45 | 33.29 ± 12.89 | 19.00 ± 0.98 | 32.12 ± 4.10 |
| campaign | 27.91 ± 1.24 | 28.33 ± 0.08 | 29.91 ± 0.14 | 28.40 ± 0.32 | 15.80 ± 0.13 | 28.68 ± 0.21 | 35.44 ± 0.07 | 36.84 ± 0.06 | 13.05 ± 4.47 | 14.51 ± 2.47 | 35.21 ± 0.82 | 85.00 ± 9.13 | 86.67 ± 7.45 | 18.35 ± 6.15 | 37.62 ± 0.74 | 29.90 ± 0.96 | 32.12 ± 1.10 |
| cardio | 55.88 ± 4.43 | 53.57 ± 0.67 | 40.17 ± 1.51 | 60.87 ± 0.73 | 15.89 ± 1.81 | 48.23 ± 1.68 | 56.68 ± 0.74 | 57.59 ± 0.51 | 42.78 ± 10.47 | 16.09 ± 1.04 | 45.80 ± 0.86 | 77.33 ± 24.28 | 70.67 ± 17.97 | 18.35 ± 6.15 | 37.62 ± 0.74 | 27.84 ± 5.61 | 26.80 ± 1.87 |
| cardiotocography | 43.62 ± 2.11 | 40.83 ± 0.26 | 32.37 ± 0.29 | 46.20 ± 1.18 | 27.15 ± 0.91 | 33.53 ± 1.56 | 50.23 ± 0.37 | 49.29 ± 2.63 | 46.28 ± 1.30 | 27.64 ± 0.53 | 36.10 ± 0.67 | 3.74 ± 0.13 | 3.67 ± 0.11 | 31.16 ± 0.49 | 33.84 ± 3.20 | 27.55 ± 1.23 | |
| celeba | 6.26 ± 0.41 | 10.28 ± 0.48 | 6.07 ± 0.27 | 11.19 ± 0.62 | 1.81 ± 0.02 | 6.88 ± 2.06 | 9.53 ± 0.55 | 9.28 ± 0.59 | 4.65 ± 3.19 | 2.37 ± 0.28 | 8.95 ± 0.56 | 25.77 ± 4.08 | 29.41 ± 6.38 | 5.77 ± 1.59 | 5.19 ± 0.23 | 9.25 ± 1.47 | 7.68 ± 0.63 |
| census | 7.30 ± 0.49 | 8.52 ± 0.23 | 8.82 ± 0.09 | 8.66 ± 0.23 | 6.87 ± 0.23 | 8.75 ± 0.28 | 6.23 ± 0.16 | 6.23 ± 0.16 | 6.52 ± 2.72 | 6.11 ± 0.18 | 7.30 ± 0.19 | 46.56 ± 5.17 | 37.43 ± 5.18 | 8.34 ± 0.92 | 9.00 ± 0.09 | 8.56 ± 0.23 | 8.09 ± 0.30 |
| Average | 37.47 | 36.03 | 33.83 | 36.60 | 18.92 | 35.20 | 34.34 | 33.34 | 28.97 | 18.55 | 34.40 | 39.60 | 39.95 | 23.62 | 32.24 | 33.25 | 31.89 |
| 20news | 6.24 ± 0.36 | 6.38 ± 0.44 | 6.90 ± 0.36 | 6.24 ± 0.23 | 6.88 ± 0.42 | 6.17 ± 0.12 | 6.09 ± 0.29 | 6.23 ± 1.07 | 8.71 ± 0.87 | 6.25 ± 0.21 | 7.16 ± 0.49 | 6.25 ± 0.21 | 6.44 ± 0.58 | 6.86 ± 3.01 | 8.24 ± 1.48 | | |
| agnews | 6.36 ± 0.22 | 6.78 ± 0.07 | 8.16 ± 0.03 | 6.11 ± 0.02 | 12.47 ± 0.61 | 7.24 ± 0.07 | 5.76 ± 0.01 | 5.65 ± 0.01 | 6.42 ± 0.58 | 12.51 ± 0.62 | 5.87 ± 0.02 | 6.26 ± 1.28 | 8.45 ± 0.06 | 6.17 ± 0.91 | 7.78 ± 1.07 | 7.55 ± 1.04 | 7.77 ± 2.43 |
| amazon | 5.83 ± 0.09 | 5.89 ± 0.01 | 6.22 ± 0.01 | 5.69 ± 0.02 | 5.79 ± 0.13 | 6.06 ± 0.06 | 5.50 ± 0.01 | 5.96 ± 0.02 | 5.44 ± 0.43 | 5.80 ± 0.11 | 5.87 ± 0.02 | 5.49 ± 0.40 | 6.22 ± 0.08 | 5.71 ± 0.01 | 5.72 ± 0.54 | 5.80 ± 0.34 | 5.59 ± 0.61 |
| imdb | 4.68 ± 0.04 | 4.69 ± 0.09 | 4.83 ± 0.02 | 4.59 ± 0.01 | 4.88 ± 0.06 | 4.74 ± 0.03 | 4.48 ± 0.01 | 4.96 ± 0.03 | 4.38 ± 0.28 | 4.87 ± 0.04 | 4.74 ± 0.01 | 4.74 ± 0.52 | 4.69 ± 0.03 | 4.59 ± 0.03 | 4.77 ± 0.34 | 4.80 ± 0.30 | |
| yelp | 6.96 ± 0.05 | 7.29 ± 0.01 | 8.27 ± 0.05 | 6.88 ± 0.03 | 8.52 ± 0.18 | 7.31 ± 0.17 | 6.47 ± 0.01 | 7.24 ± 0.03 | 6.67 ± 0.60 | 8.52 ± 0.19 | 7.04 ± 0.01 | 5.42 ± 0.93 | 8.50 ± 0.13 | 6.92 ± 0.01 | 6.59 ± 0.78 | 7.56 ± 0.39 | 6.84 ± 0.92 |
| Average | 6.01 | 6.21 | 6.84 | 5.90 | 8.08 | 6.40 | 5.68 | 6.02 | 5.87 | 8.08 | 5.91 | 5.58 | 7.00 | 5.93 | 6.27 | 6.55 | 6.43 |

## A.2 Tennessee Eastman Process

The TE process was first proposed by Downs & Vogel (1993), as a large-scale chemical process for benchmarking control algorithms, as well as fault detection and diagnosis methods. We follow the description of Reinartz et al. (2021). This chemical process consists of 4 chemical reactions, for the production of 2 liquid liquid components, denoted $G$ and $E$, and 4 gaseous reactants, denoted $A$, $B$, $C$ and $D$. The reactions are as follows,

$$
\begin{aligned}
A(\text{g}) + C(\text{g}) + D(\text{g}) &\rightarrow G(\text{liq}) \quad \text{Product 1,} \\
A(\text{g}) + C(\text{g}) + E(\text{g}) &\rightarrow H(\text{liq}) \quad \text{Product 2,} \\
A(\text{g}) + E(\text{g}) &\rightarrow F(\text{liq}) \quad \text{Byproduct,} \\
3D(\text{g}) &\rightarrow 2F(\text{liq}) \quad \text{Byproduct.}
\end{aligned}
\tag{12}
$$

The chemical plant is composed by 5 processing units: reactor, product condenser, vapor-liquid separator, recycle compressor and product stripper. We refer readers to Reinartz et al. (2021) and Montesuma et al. (2024b) for further details on how these components work together in the system. As described by Downs & Vogel (1993), based on equation 12, there are 6 different *modes of operation*, corresponding to different product mass ratio and product rate. We detail these in Table 3.

Table 3: TE process operation modes in terms of $G/H$ mass ratio and production rate.

| Mode | Mass Ratio | Production rate |
|------|------------|-----------------|
| 1 | 50/50 | 7038 kg h$^{-1}$ G and 7038 kg h$^{-1}$ H |
| 2 | 10/90 | 1408 kg h$^{-1}$ G and 12,669 kg h$^{-1}$ H |
| 3 | 90/10 | 10,000 kg h$^{-1}$ G and 1111 kg h$^{-1}$ H |
| 4 | 50/50 | maximum production rate |
| 5 | 10/90 | maximum production rate |
| 6 | 90/10 | maximum production rate |

From the chemical plant that performs the equations in 12, there are 54 variables divided into measured (XME) and manipulated (XMV). We show these in Table 4. Out of the 54 variables, we do as in Reinartz et al. (2021), that is, we use a subset of 34 variables, boldened in Table 4. In the simulations of Reinartz et al. (2021), the TE process is simulated for 100 hours, with a sampling rate of 3 minutes. As a result, each simulation is represented as a time series $\mathbf{x} \in \mathbb{R}^{600 \times 34}$, where $\mathbf{x}(t) \in \mathbb{R}^{34}$.

Table 4: Process variables used in the TE process, divided into measurements (XME) and manipulated (XMV). Bold rows indicate that the variable is used in our experiments.

| Variable | Description | Variable | Description | Variable | Description | Variable | Description |
|----------|-------------|----------|-------------|----------|-------------|----------|-------------|
| **XME(1)** | **A Feed (kscmh)** | **XME(15)** | **Stripper Level (%)** | XME(29) | Component A in Purge (mol %) | **XMV(2)** | **E Feed (%)** |
| **XME(2)** | **D Feed (kg/h)** | **XME(16)** | **Stripper Pressure (kPa gauge)** | XME(30) | Component B in Purge (mol %) | **XMV(3)** | **A Feed (%)** |
| **XME(3)** | **E Feed (kg/h)** | **XME(17)** | **Stripper Underflow (m³/h)** | XME(31) | Component C in Purge (mol %) | **XMV(4)** | **A & C Feed (%)** |
| **XME(4)** | **A & C Feed (kg/h)** | **XME(18)** | **Stripper Temp (°C)** | XME(32) | Component D in Purge (mol %) | **XMV(5)** | **Compressor recycle valve (%)** |
| **XME(5)** | **Recycle Flow (kscmh)** | **XME(19)** | **Stripper Steam Flow (kg/h)** | XME(33) | Component E in Purge (mol %) | **XMV(6)** | **Purge valve (%)** |
| **XME(6)** | **Reactor Feed rate (kscmh)** | **XME(20)** | **Compressor Work (kW)** | XME(34) | Component F in Purge (mol %) | **XMV(7)** | **Separator liquid flow (%)** |
| **XME(7)** | **Reactor Pressure (kscmh)** | **XME(21)** | **Reactor Coolant Temp (°C)** | XME(35) | Component G in Purge (mol %) | **XMV(8)** | **Stripper liquid flow (%)** |
| **XME(8)** | **Reactor Level (%)** | **XME(22)** | **Separator Coolant Temp (°C)** | XME(36) | Component H in Purge (mol %) | **XMV(9)** | **Stripper steam valve (%)** |
| **XME(9)** | **Reactor Temperature (°C)** | XME(23) | Component A to Reactor (mol %) | XME(37) | Component D in Product (mol %) | **XMV(10)** | **Reactor coolant (%)** |
| **XME(10)** | **Purge Rate (kscmh)** | XME(24) | Component B to Reactor (mol %) | XME(38) | Component E in Product (mol %) | **XMV(11)** | **Condenser Coolant (%)** |
| **XME(11)** | **Product Sep Temp (°C)** | XME(25) | Component C to Reactor (mol %) | XME(39) | Component F in Product (mol %) | **XMV(12)** | **Agitator Speed (%)** |
| **XME(12)** | **Product Sep Level (%)** | XME(26) | Component D to Reactor (mol %) | XME(40) | Component G in Product (mol %) | | |
| **XME(13)** | **Product Sep Pressure (kPa gauge)** | XME(27) | Component E to Reactor (mol %) | XME(41) | Component H in Product (mol %) | | |
| **XME(14)** | **Product Sep Underflow (m³/h)** | XME(28) | Component F to Reactor (mol %) | **XMV(1)** | **D Feed (%)** | | |

The simulations in Reinartz et al. (2021) are faulty or normal. For faulty simulations, a different fault type is introduced between 28 possible categories at $t_0 = 30h$. We refer readers to the original paper for more information about the different categories. In this paper, we construct 6 different AD datasets, one for each mode of operation. On each mode of operation, we use 100 normal simulations, and 1 faulty simulation for each fault type. This process results in 128 time series per mode of operation.

For applying AD algorithm, we further pre-process the time series obtained from the simulation of the TE process. We decompose each time series in windows of 20 hours, i.e., 60 samples. Within each window, we

compute the mean and standard-deviation per sensor, that is,

$$\mu_{tj} = \frac{1}{60} \sum_{i=t}^{t+60} x_{ij}, \text{ and, } \sigma_{tj} = \sqrt{\frac{1}{59} \sum_{i=t}^{t+60} (x_{ij} - \mu_{tj})^2},$$

where $t$ is the index of the window. We use the concatenation $(\mu_t, \sigma_t) \in \mathbb{R}^{68}$ as the representation for the $t-$th window. As a result of this process, we extract 30 windows for each time series. This results on 6 datasets with $128 \times 30 = 3840$ samples each. We then proceed to apply AD algorithms at the level of windows. Here, we note an important detail. For OT algorithms, we compare samples based on the Euclidean distance. Computing this distance over $\mathbf{x} = (\mu, \sigma)$ and $\mathbf{y} = (\mu', \sigma')$ boils down to,

$$\|\mathbf{x}_i - \mathbf{x}_j\|_2^2 = \|\mu - \mu'\|_2 + \|\sigma - \sigma'\|_2,$$

which is the squared $2-$Wasserstein distance $W_2(P, Q)^2$ between axis-aligned Gaussians $P = \mathcal{N}(\mu, \sigma\mathbf{I})$ and $Q = \mathcal{N}(\mu', \sigma'\mathbf{I})$.

### A.2.1 Results per anomaly percentage

In Figure 14, we show an overview of the AUC-ROC of different methods per percentage of anomalies. Overall, our MROT outperforms OT with the regularized Coulomb cost on all modes. Furthermore, our method outperforms or stays competitive with other methods on most modes, such as 1, 3 and 6.

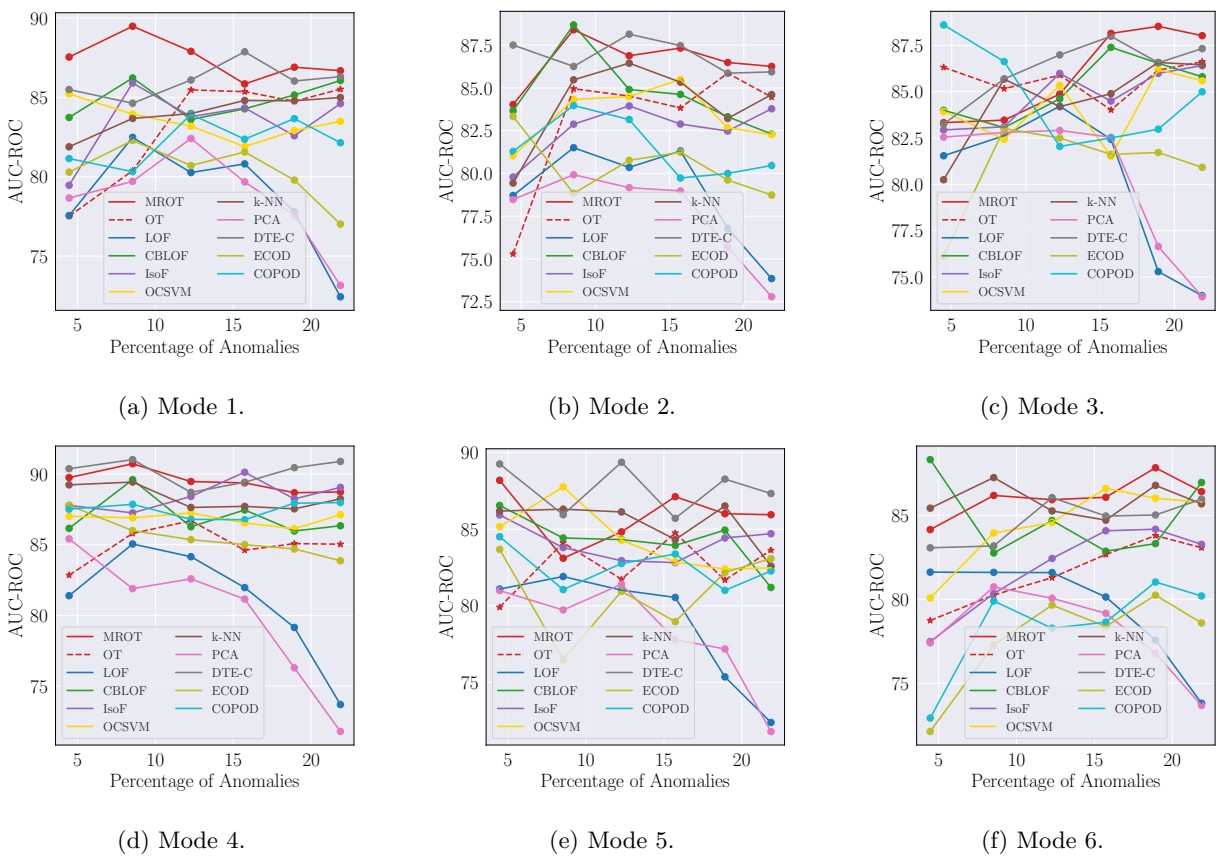

Figure 14: AUC-ROC per percentage of anomalies on the TE benchmark.

