# OpenReview forum: "Unsupervised Anomaly Detection through Mass Repulsing Optimal Transport"
_TMLR — Accepted by TMLR_

### Review · Reviewer_aamh · 2025-03-21

**Summary Of Contributions:**

This paper introduces Mass Repulsing Optimal Transport (MROT), a novel approach to anomaly detection that modifies Optimal Transport (OT) by enforcing mass displacement while minimizing effort. Anomalies, located in low-density regions, incur higher transportation costs, while inliers shift minimally. This formulation yields a new anomaly score, demonstrating superior performance over existing methods in benchmark and fault detection tasks.

**Audience:**

Yes

**Claims And Evidence:**

Yes

**Requested Changes:**

Please see weaknesses.

**Strengths And Weaknesses:**

Strengths

- The application of OT for AD is an interesting and promising direction.
- The concept of using repulsive cost to identify anomalies sounds novel.

Weaknesses

- The literature on AD is huge, but the related works seem to overlook many important AD studies, particularly DNN-based approaches such as GAN-based and AutoEncoder-based AD, which are widely used in industry. Most of the discussed related works focus on traditional ML methods. I suggest providing a more comprehensive discussion of the AD literature. While I understand that the authors primarily focus on OT, the AD problem itself is the main target. Regardless of whether OT is used, it is essential to consider other AD techniques.

- Due to the limited discussion on modern DNN-based AD methods, the experiments appear relatively weak, as there is insufficient evidence to demonstrate that the proposed method is truly practical and competitive with state-of-the-art AD techniques.

- When using a repulsive cost, is there any difficulty in solving the OT problem? It seems that a similar approach to computing Sinkhorn could be applied. If not, the technical novelty may be relatively limited, although the conceptual contribution is interesting. Please kindly provide more clarifications.

---

> ### Author Response · Authors · 2025-04-19
> **Rebuttal**
>
> Dear Reviewer,
>
> Thank you for your insightful review. We are pleased that you found the application of OT for AD to be __interesting__ and __promising__, and that our repulsive cost is __novel__. In the following, we address the weakness/request changes section of your review.
>
> > The literature on AD is huge, but the related works seem to overlook many important AD studies, particularly DNN-based approaches such as GAN-based and AutoEncoder-based AD, which are widely used in industry. Most of the discussed related works focus on traditional ML methods. I suggest providing a more comprehensive discussion of the AD literature. While I understand that the authors primarily focus on OT, the AD problem itself is the main target. Regardless of whether OT is used, it is essential to consider other AD techniques.
>
> Thank you for suggesting additional discussions in our paper. Indeed, we can extend the discussion of AD literature to deep learning-based methods. Please see see our extended discussion of DNN-based AD methods at the end of our rebuttal. You may find the additional discussion in the revised version of our manuscript, especially Section 2 Related Work. If you find any specific work missing from the updated discussion, please, let us know.
>
> > Due to the limited discussion on modern DNN-based AD methods, the experiments appear relatively weak, as there is insufficient evidence to demonstrate that the proposed method is truly practical and competitive with state-of-the-art AD techniques.
>
> In our experiments section, we compared our method to different strategies in the state-of-the-art. For instance, we included some classical algorithms such as OCSVM, proposed in 1999, and recent methods such as ECOD (TPAMI 2022) and DTE (ICLR 2024). In total, we evaluate 17 algorithms on the AdBench benchmark, and 11 algorithms on the TEP benchmark. If you feel that any particular relevant algorithm is missing from our empirical comparison, please let us know
>
> > When using a repulsive cost, is there any difficulty in solving the OT problem? It seems that a similar approach to computing Sinkhorn could be applied. If not, the technical novelty may be relatively limited, although the conceptual contribution is interesting. Please kindly provide more clarifications.
>
> A key advantage is that off‑the‑shelf OT solvers readily handle our formulation. As a result, we can solve our MROT problem using off-the-shelf OT solvers, such as the Sinkhorn algorithm (entropic regularized problem) or linear programming (exact MROT). The same applies for the underlying theory, as we can use classical OT theory for analyzing our problem.
>
> We stress that the novelty of our approach resides on the __cost engineering__ ideas we introduce. While the underlying OT problem is solved with existing tools, we do not claim innovations on this area.
>
> # Additional discussion on deep learning-based methods
>
> Deep Learning-based Anomaly Detection. As Pang et al. (2021) reviews, deep learning is used for AD in mainly 3 ways: for i) learning powerful feature extractors, ii) learning representations of normality, iii) building an end-to-end anomaly score. Besides iii, i and ii allow for the synergy between deep and traditional AD algorithms. Indeed, one can use off-the-shelf AD strategies over the latent space of a neural net. We explore this possibility in Section 4.1.
>
> In addition to this possibility, we give a few examples of how deep learning has contributed to AD. For example, Ruff et al. (2018) proposed a strategy for performing deep one class classification, which echoes the ideas of One Class SVM (OCSVM) Schölkopf et al. (1999).
>
> Furthermore, generative modeling has deeply contributed to AD. For instance, generative adversarial nets (Goodfellow et al., 2014) can serve as a modeling step for capturing features of the underlying distribution P . For instance, in Schlegl et al. (2019) and Zenati et al. (2018), the authors combine reconstruction errors with the distance of the predictions of the discriminator for building an anomaly score. Akcay et al. (2018) takes this idea further. This strategy creates a encoder-decoder-encoder architecture for the generator network. The model is trained to model normal images, by mapping them to the latent space, reconstructing back the image, then mapping again to the latent space. Anomalous images are then detected via the usual reconstruction loss, plus the distance between latent representations and the difference between discriminator’s predictions. Another direction consists of using variational inference Dias et al. (2020) or diffusion models Livernoche et al. (2024).

---

> ### Author Response · Authors · 2025-04-19
> **Rebuttal - Additional References**
>
> # Additional References
>
> Thomas Schlegl, Philipp Seeböck, Sebastian M Waldstein, Georg Langs, and Ursula Schmidt-Erfurth. f-anogan: Fast unsupervised anomaly detection with generative adversarial networks. Medical image analysis, 54:30–44, 2019.
>
> Houssam Zenati, Chuan Sheng Foo, Bruno Lecouat, Gaurav Manek, and Vijay Ramaseshan Chandrasekhar. Efficient gan-based anomaly detection. arXiv preprint arXiv:1802.06222, 2018.
>
> Samet Akcay, Amir Atapour-Abarghouei, and Toby P Breckon. Ganomaly: Semi-supervised anomaly detection via adversarial training. In Asian conference on computer vision, pp. 622–637. Springer, 2018.
>
> Lukas Ruff, Robert Vandermeulen, Nico Goernitz, Lucas Deecke, Shoaib Ahmed Siddiqui, Alexander Binder, Emmanuel Müller, and Marius Kloft. Deep one-class classification. In International conference on machine learning, pp. 4393–4402. PMLR, 2018.

---

### Review · Reviewer_8CYE · 2025-03-23

**Summary Of Contributions:**

This paper introduces a novel anomaly detection framework by reformulating optimal transport (OT) into what the authors call Mass Repulsing Optimal Transport (MROT). The key idea is to modify the ground cost so that samples are forced to “repel” their own mass away from a predefined exclusion zone (typically defined by k‐nearest neighbors). By doing so, the method accentuates the differences between normal and anomalous points—since anomalies, often lying in sparse regions, incur a higher transportation cost. The paper also provides a principled way to transform these costs into interpretable anomaly scores, and demonstrates the efficacy of the approach across several benchmarks including AdBench and the Tennessee Eastman process.

**Audience:**

Yes

**Claims And Evidence:**

Yes

**Requested Changes:**

I do not have any suggested adjustments.

**Strengths And Weaknesses:**

Strengths:

Innovative Approach: The engineered ground cost and the formulation of MROT are quite original. The idea of using mass repulsion to highlight anomalies is refreshing and mathematically interesting.

Comprehensive Experiments: The paper presents extensive empirical evaluations over multiple benchmarks, comparing with state-of-the-art methods. The inclusion of both classical and modern techniques adds weight to the experimental claims.

Interpretability: The use of a CDF transformation for the anomaly score helps in making the outputs more understandable and actionable, which is much appreciated in practical deployments.



Weaknesses:

Scalability Concerns: As the authors themselves note, OT’s O(n²) storage and O(n³ log n) computational complexities may limit the practical use of the method on very large datasets without further approximation or batching strategies.

High-Dimensional Data: The method seems to struggle with high-dimensional features (as seen in some NLP datasets), and while this is not uncommon in OT-based approaches, it remains a limitation.

Regression Step Ambiguity: The process to generalize the anomaly score via regression could use further clarification. Details about model selection, training stability, and potential overfitting are somewhat lacking.

Hyper-Parameter Sensitivity: Although a robustness analysis is provided, more insight into choosing the exclusion zone parameter (number of neighbors) and entropic regularization parameter would benefit the reader.

---

> ### Author Response · Authors · 2025-04-18
> **Rebuttal**
>
> Dear Reviewer,
>
> Thank you for your encouraging review. Especially, we are glad that you found our approach __refreshing and mathematically interesting__. We invite you to take a look on our additional theoretical analysis in the manuscript, as well as in the response to reviewer __Ey1z__. In the following, we address some of the weaknesses you mentioned,
>
> > Regression Step Ambiguity: The process to generalize the anomaly score via regression could use further clarification. Details about model selection, training stability, and potential overfitting are somewhat lacking.
>
> Indeed, in the first version of our manuscript we did not include this discussion. We refer the reviewer to Section 4.3 in our revised manuscript which now includes an ablation on the choice of regression algorithm, and a discussion on model selection.
>
> In a nutshell, given a training dataset for the regression algorithm, $\set{\mathbf{x}\_{i}^{(P)},\hat{F}\_{\mathcal{T}}(t\_{i})}\_{i=1}^{n}$, we can employ off-the-shelf validation methods, such as k-fold cross validation. For simplicity, we can imagine a model selection process which partitions this set into a training and validation. The parameters of the regression algorithm can then be chosen based on the error on the validation set.
>
> > Hyper-Parameter Sensitivity: Although a robustness analysis is provided, more insight into choosing the exclusion zone parameter (number of neighbors) and entropic regularization parameter would benefit the reader.
>
> Besides the robustness analysis, we emphasize the following guidelines for choosing the number of neighbors in the exclusion zone,
>
> _"As a general guideline, it is better to limit the number of nearest neighbors, as anomalous examples are likely rare. As a consequence, using a high value for k may lead to the inclusion of normal points in the neighborhood of anomalous ones."_
>
> Conversely, for $\epsilon$, we give an additional analysis. Note that, from Figure 12 (revised manuscript), there is a performance degradation with increasing $\epsilon$. From classical OT theory, using $\epsilon > 0$ leads to a non-sparse OT plan. This means that there are more indices $(i, j)$, for which $\gamma\_{ij} > 0$. Likewise, if we let $\epsilon \rightarrow +\infty$, then $\gamma$ converges to the trivial coupling $\gamma\_{ij} = p\_{i}p\_{j}$. Assuming uniform importances, i.e., $p\_{i} = n^{-1}$, and plugging these results back into equation 10, we have,
>
> $$t_{i} = \dfrac{1}{n}\sum_{j=1}^{n}\tilde{c}_{k}(x\_{i}^{(P)}, x\_{j}^{(P)})$$
>
> that is, for a large entropic penalty, we end up simply computing the average cost of each sample. The implication of this analysis is as follows: the MROT plan captures some structure in the relationship between anomalous and normal points, which ends up contributing to a more accurate anomaly score.

---

### Review · Reviewer_Ey1z · 2025-04-08

**Summary Of Contributions:**

This paper introduces a novel approach to anomaly detection called Mass Repulsing Optimal Transport (MROT), which modifies classical Optimal Transport (OT) by forcing samples to redistribute their mass away from their original positions while minimizing transportation cost.  The paper's innovation and experimental design are commendable, yet there is room for improvement in methodological details, theoretical analysis, and computational efficiency.

**Audience:**

Yes

**Broader Impact Concerns:**

See contributions.

**Claims And Evidence:**

Yes

**Requested Changes:**

See weaknesses.

**Strengths And Weaknesses:**

Strengths:
1. Innovation: A novel OT problem (MROT) is proposed, which achieves anomaly detection through a repulsive cost function. This idea is original in the field of OT. The combination of OT theory and anomaly detection problems provides a new perspective for unsupervised anomaly detection.
2. Experimental Design: The experiments cover a variety of benchmark datasets (such as AdBench), demonstrating the universality and practicality of the method.

Weakness:
1. Clarity of Method Details The core ideas of MROT (such as the design of the repulsive cost function and the definition of the "repulsion region") are described rather abstractly in the text, lacking intuitive mathematical or geometric explanations.
2. Depth of Theoretical Analysis: There is a lack of discussion on the theoretical properties of MROT (such as convergence and uniqueness), especially in comparison with classical OT problems.
3. The resolution of Figures 4-7 is low. It is recommended to provide clearer vector graphics.
The labels on the bar chart in Figure 8 are too dense. They can be simplified or displayed separately.
References: The citation format of some references is not uniform (such as "Montesuma et al. ()" missing the year).

---

> ### Author Response · Authors · 2025-04-18
> **Rebuttal**
>
> Dear Reviewer,
>
> Thank you for your insightful review. In the following, we comment on your requested changes.
>
> > Clarity of Method Details The core ideas of MROT (such as the design of the repulsive cost function and the definition of the "repulsion region") are described rather abstractly in the text, lacking intuitive mathematical or geometric explanations.
>
> In our revised manuscript, we added the following remark after eq. 6
>
> Assume $x_0 \in \mathcal{X}$ fixed. Our cost engineering strategy replaces the base ground-cost $c(x_0, y)$, in the neighborhood $\mathcal{N}(x_0)$ with a value $L$. Assume, for argumentation purposes, that $L \rightarrow +\infty$. This choice means that it is infinitely costly for each point to keep its own mass. As a result, samples are encouraged to send their mass to the immediate outside of their neighborhood  -- thus, we call this idea \emph{mass repulsive} OT.
>
> To ground some intuition, we show, in Figure 3, a comparison using the squared Euclidean distance as the ground-cost, i.e., $c(x, y) = || x - y ||\_{2}\^{2}$. Alongside the base cost, we show our engineered cost, and the repulsive cost $(1 + c(x, y))^{-1}$. Here, we define the neighborhood of $x_0$ through the $\rho-$closed ball centered at $x_0$, i.e., $\mathcal{N}(x_0) = \{y : \mathbb{R}\^{2}:|| x - y ||\_{2} \leq \rho\}$. Likewise, we use $\rho = 1$, and, $L = \text{sup}\_{y \in \mathcal{N}(x\_0)}c(x, y)$. This particular choice leads to nice theoretical properties for the continuous MROT problem (see equation 7 below).
>
> Note that, due the particular choices we made, the supremum takes its value on the frontier $\partial \mathcal{N}(x_0) = \{y : \mathbb{R}^{2}: || x_0 - y ||_{2} = \rho\}$, in which case $c(x_0, y) = \rho^{2}$, independently of $x_0$. This choice for $L$ effectively flattens the ground-cost around the vicinity of $x_0$. As we show in the next section, this is essential for ensuring the existence of the continuous MROT problem. In practice, it might be beneficial to set $L$ to a large, finite constant to ensure points are transported outside $\mathcal{N}(x_0)$. Note that, as we remark in the next section, this does not harm the existence of the \textbf{discrete} MROT solution.
>
> > Depth of Theoretical Analysis: There is a lack of discussion on the theoretical properties of MROT (such as convergence and uniqueness), especially in comparison with classical OT problems.
>
> Thank you for this question. Indeed, the tools for analyzing the MROT problem are very similar to OT problems, after all, we are still solving an optimal transportation problem. We added a new section discussing the existence and uniqueness of MROT plans. We invite the reviewer to read section 3.3 in our revised manuscript for further details.
>
> In a nutshell, the conditions for the existence of the continuous problem depend on the continuity of the underlying ground-cost. Assuming one chooses a continuous base cost (such as the squared Euclidean distance we use in the main paper), one needs to carefully choose the lower bound $L$ to keep the continuity of $\tilde{c}$. As we show in the paper, choosing $L = \text{sup}\_{y \in \mathcal{N}(x)}c(x,y)$ provides an adaptative choice that keeps the contiuity, and thus existence, of the continuous MROT problem. Concerning uniqueness, MROT plans are, in general, non-unique, since they do not suffice the conditions described in (Figalli and Gaudo, 2021, Section 2.7.1).
>
> As we coment on the paper, the case for the discrete MROT problem is easier to analyze. While, like classical OT, we cannot guarantee uniqueness of a solution, existence is straightforward as $\Gamma(\mathbf{p},\mathbf{p})$ is non-empty and compact, and the objective function is continuous. These arguments do not depend on the base cost $c$, nor the neighborhood function $\mathcal{N}$. In practice, we found that using $k-$nearest neighbors for the definition of neighborhood and setting $C\_{i\ell} = \text{max}\_{j=1,\cdots,n}C\_{ij}$, where $\ell \in \mathcal{N}(\mathbf{x}\_{i}\^{(P)})$ works better.
>
> > The resolution of Figures 4-7 is low. It is recommended to provide clearer vector graphics. The labels on the bar chart in Figure 8 are too dense. They can be simplified or displayed separately.
>
> Figures 4 through 7 were generated using matplotlib with vector graphics (figures were saved in .pdf format with tex fonts). For better visualizations, we increased the size of these figures. For figure 7, we changed the `pcolormesh` plot in favor of `contourf` to reduce the artifacts. We also simplified the labels of Figure 8.
>
> > References: The citation format of some references is not uniform (such as "Montesuma et al. ()" missing the year).
>
> We searched the .pdf for the missing year in the citation but could not find it. Could the reviewer kindly give a pointer in the .pdf to where we can find the ref with a missing year?

---

> > ### Comment · Reviewer_Ey1z · 2025-05-09
> >
> > The author has provided clear and sufficient responses to my concerns.

---

### Author Response · Authors · 2025-04-19
**Summary of Changes**

Dear Reviewers,

Thank you for your thoughtful feedback on our manuscript. We uploaded a revised version, which includes these changes,

Reviewer __Ey1z__
- We included __Remark 3.1__ about the geometric intuition behind our cost engineering process.
- We included __Section 3.3__ about the theoretical analysis of the continuous and discrete MROT problems, analyzing existence and uniqueness of MROT plans.
- We revised the generation of Figures 4 through 7. When applicable, we enlarged the figures for better visualization. We enlarged the labels of Figure 8, and removed the redundancies.

Reviewer __8CYE__

We added a few more discussions on __Section 4.3 Ablations__

- We added a discussion on the choice of $\epsilon$, and the effect of large entropic regularization.
- We included a discussion, alongside an additional figure, about the robustness of our algorithm with respect the choice of regression algorithm.
- We included a discussion about model selection.

Reviewer __aamh__
- We included additional discussion on the related works section about DNN-based AD. We also included 4 references to works using these principles, besides the 2 we had already covered in the first version of our manuscript (Dias et al., 2020; Livernoche et al., 2024)

# Additional References

Thomas Schlegl, Philipp Seeböck, Sebastian M Waldstein, Georg Langs, and Ursula Schmidt-Erfurth. f-anogan: Fast unsupervised anomaly detection with generative adversarial networks. Medical image analysis, 54:30–44, 2019.

Houssam Zenati, Chuan Sheng Foo, Bruno Lecouat, Gaurav Manek, and Vijay Ramaseshan Chandrasekhar. Efficient gan-based anomaly detection. arXiv preprint arXiv:1802.06222, 2018.

Samet Akcay, Amir Atapour-Abarghouei, and Toby P Breckon. Ganomaly: Semi-supervised anomaly detection via adversarial training. In Asian conference on computer vision, pp. 622–637. Springer, 2018.

Lukas Ruff, Robert Vandermeulen, Nico Goernitz, Lucas Deecke, Shoaib Ahmed Siddiqui, Alexander Binder, Emmanuel Müller, and Marius Kloft. Deep one-class classification. In International conference on machine learning, pp. 4393–4402. PMLR, 2018.

---

### Decision · Action_Editor_S442 · 2025-05-19

**Recommendation:** Accept with minor revision

**Comment:**

The reviewers agree that the proposed method is novel and interesting, and the authors have adequately responded to most of the reviewers' concerns. The remaining concern that an in-depth comparison with neural anomaly detection methods is missing is valid, but in the light of the editorial policy of TMLR (as I understand it) it does not pregnant publication.

I suggest some minor editorial changes: The text had occasional grammar mistakes and typos, probably due to editing. Similarly, the caption off figure 5 refers to (a), which is probably figure 4. I suggest to carefully read the paper before submitting final files. Furthermore, in figure 3, the engineered cost is not largest for points close to $x_0$. I suggest redrawing this figure for a larger $L$, or adjusting the claim in the text (if I understood the claim correctly).

**Audience:**

Yes. Anomaly detection remains a relevant task in machine learning and data mining, and the paper presents a nontrivial advance in this field.

**Claims And Evidence:**

While since reviewers commented on the fact that some state of the art anomaly detection approaches were not compared to, there seems to be consensus that there is sufficient evidence validating the provided approach.